**COMMUNICATIONS**

# Transancestral mapping and genetic load in systemic lupus erythematosus

Carl D. Langefeld et al.[#]

Systemic lupus erythematosus (SLE) is an autoimmune disease with marked gender and ethnic disparities. We report a large transancestral association study of SLE using Immunochip genotype data from 27,574 individuals of European (EA), African (AA) and Hispanic Amerindian (HA) ancestry. We identify 58 distinct non-HLA regions in EA, 9 in AA and 16 in HA ($\sim$50% of these regions have multiple independent associations); these include 24 novel SLE regions ($P < 5 \times 10^{-8}$), refined association signals in established regions, extended associations to additional ancestries, and a disentangled complex HLA multigenic effect. The risk allele count (genetic load) exhibits an accelerating pattern of SLE risk, leading us to posit a cumulative hit hypothesis for autoimmune disease. Comparing results across the three ancestries identifies both ancestry-dependent and ancestry-independent contributions to SLE risk. Our results are consistent with the unique and complex histories of the populations sampled, and collectively help clarify the genetic architecture and ethnic disparities in SLE.

#A full list of authors and their affiliations appears at the end of the paper.

Systemic lupus erythematosus (SLE) (OMIM 152,700) is a chronic autoimmune disease that affects multiple organs, and disproportionately affects women and individuals of non-European ancestry[1]. Candidate gene and genome-wide association studies[2–4] have successfully identified ~90 SLE risk loci that explain a significant proportion of SLE's heritability[5–8]. These studies have been largely restricted to populations of European ancestry (EA). Yet, much of the heritability of SLE risk remains unexplained in EA populations, and is largely unknown in other ancestries. Here, we report the results of genotyping large samples of individuals of EA, African American (AA) and Hispanic (Amerindian) American ancestry (HA) on the Illumina Infinium Immunochip (196,524 polymorphisms: 718 small insertion deletions, 195,806 single nucleotide polymorphisms (SNPs)), a microarray designed to perform both deep replication and fine mapping of established major autoimmune and inflammatory disease loci[9].

This study identifies 58 distinct non-HLA regions in EA, 9 in AA and 16 in HA. Approximately 50% of the associated regions have multiple independent associations. These 58 regions include 24 novel SLE regions reaching genome-wide significance ($P < 5 \times 10^{-8}$). Further, these results localize the association signals in established regions and extended associations to additional ancestries (for example, EA to AA or HA). Adjusting for the associated HLA alleles disentangles a complex multigenic effect just outside of the HLA region. The association between SLE and the risk allele genetic load (risk allele count) exhibits an accelerating nonlinear trend, greater than expected if the loci were acting independently on risk. This nonlinear risk relationship leads us to posit a cumulative hit hypothesis for autoimmune disease. Finally, we report both ancestry-dependent and ancestry-independent contributions to SLE risk.

## Results

**SLE genetic association study.** In total, 27,574 SLE cases and controls from three ancestral groups were genotyped and passed quality control for the Immunochip (AA: 2,970 cases, 2,452 controls; EA: 6,748 cases, 11,516 controls; HA: 1,872 cases and 2,016 controls). Altogether, 146,111 SNPs passed quality control analyses in at least one ancestry (AA: 128,385, EA: 120,873, HA: 120,786). Restricting linkage disequilibrium (LD) to $r^2 < 0.2$ yielded 46,774 uncorrelated SNPs (union across ancestries) for an estimate of the number of independent tests. To minimize ancestry-specific inflation factors, 3, 4 and 2 admixture factors were included as covariates in the logistic regression model

for the EA, AA and HA association analyses, respectively (Supplementary Fig. 1). Inflation factors, scaled to 1,000 cases and 1,000 controls, were $\lambda_{AA,1000} = 1.03$, $\lambda_{EA,1000} = 1.03$ and $\lambda_{HA,1000} = 1.13$ (Supplementary Fig. 2). Power analyses are reported in Supplementary Fig. 3.

**Single SNP association.** Table 1 shows the number of distinct regions (see Methods) within each ancestry that reached three tiers of statistical significance (Tier 1: $P < 5 \times 10^{-8}$, Tier 2: $5 \times 10^{-8} < P < 1 \times 10^{-6}$ and Tier 3: $P > 1 \times 10^{-6}$ and $P_{FDR} < 0.05$) and lists the number of regions with novel SLE associations. The Tier 1 and Tier 2 thresholds are intentionally more stringent than even the conservative Bonferroni method to reduce the Type 1 error rate on this immune-centric genotyping platform. In total, 5, 38 and 7 distinct non-HLA regions met the Tier 1 threshold of significance for the AA, EA and HA cohorts, respectively; and of these Tier 1 associations, 2, 9 and 2 were novel to SLE regardless of ethnicity or to SLE for a specific ethnicity. An additional 4, 20 and 9 distinct non-HLA regions met the Tier 2 threshold (Fig. 1).

**European ancestry.** Statistically, EA had the most power and 58 regions met Tier 1 or Tier 2 thresholds (Supplementary Data 2). Many are novel SLE risk regions, and others are novel for EA (Table 2). More than 50% of these regions had multiple independent SNPs contributing to the association, based on regional stepwise analyses. In total, 223 distinct associations met $P_{FDR} < 0.01$ (Tables 1 and 2, Supplementary Table 2), which included both well-established and novel associations.

Novel Tier 1 regions of SLE association in EA and the proximal genes include 4p16 (*DGKQ*), 6p22 (*SLC17A4* and *LRRC16A*), 6q23 (*OLIG3-LOC100130476*), 8p23 (*FAM86B3P*), 8q21 (*PKIA-ZC2HC1A*) and 17q25 (*GRB2*). Of the 20 EA Tier 2 associated regions, 16 appear novel to SLE.

**African American ancestry.** The AA sample was powered to detect OR = 1.1 to 1.2 at $\alpha = 1 \times 10^{-6}$. In addition to known regions in AA, novel AA regions identified include 5q33 (*PTTG1-MIR146A*), 6p21 (*UHRF1BP1-DEF6*) and 16q22 (*ZFP90*) (Tables 1 and 2; Supplementary Data 2). The 8p11 (*PLAT*) association is novel to SLE and was not observed in HA or EA as it was nearly monomorphic in both populations. The 1q25 region in AA is near the known anti-dsDNA-rs2205960 association between *TNFSF4* and *LOC100506023* in non-AA samples. The association at rs6681482 ($P = 8.11 \times 10^{-7}$, OR = 0.73) within *LOC100506023* appears independent and separated from the *TNFSF4* associations by a recombination hotspot. Three SNPs in this region met the stepwise significance threshold, but the strongest association in EA (rs2205960) was not genome-wide significant in AA (OR = 1.35, $P = 7.39 \times 10^{-4}$). The association with rs2431697 (OR = 0.76, $P = 1.27 \times 10^{-12}$) at 5q33 was previously associated with SLE and anti-dsDNA in EA, but not in AA (ref. 10).

**Hispanic ancestry.** HA samples had comparable power to the AA sample but exhibited more (nine versus four) novel associations at the Tier 1 and Tier 2 thresholds (Tables 1 and 2). Many regions had multiple independent associations, including cases of previously reported regions exhibiting additional novel loci. Novel Tier 1 regions include 14q31 (*GALC*) and 16p13 (*CLEC16A*). Novel Tier 2 regions include 3p11 (*EPHA3-PROS1*), 6p21 (*TCP11-SCUBE3*), 6q25 (*RSPH3*), 12q15 (*DYRK2-IFNG*), 12q21 (*SYT1*), 16q21 (*CSNK2A2-CCDC113*) and

### Table 1 | Number of non-HLA independent regions per significance tier and ancestry (number of novel regions in parentheses*).

| Tier and *P* value threshold | Ancestry | | |
| --- | --- | --- | --- |
| | African American | European ancestry | Hispanic ancestry |
| Tier 1: *P* value $< 5 \times 10^{-8}$ | 5 (2) | 38 (9) | 7 (2) |
| Tier 2: *P* value $< 1 \times 10^{-6}$ | 4 (2) | 20 (18) | 9 (7) |
| Tier 3: FDR *P* value[†] $< 0.01$ | 18 | 165 | 66 |
| Tier 3 Regions Only: FDR *P* values[‡] (not cumulative) | | | |
| 0.01–0.05 | 55 | 312 | 154 |
| 0.001–0.01 | 17 | 119 | 57 |
| 0.0001–0.001 | 1 | 40 | 9 |
| < 0.0001 | 0 | 6 | 0 |

*For Tier 1 and Tier 2 regions only; novel to SLE or first observed in specific ancestral cohort.
†Not cumulative.
‡FDR *P* value represents the Benjamini–Hochberg adjusted false discovery rate *P* value.

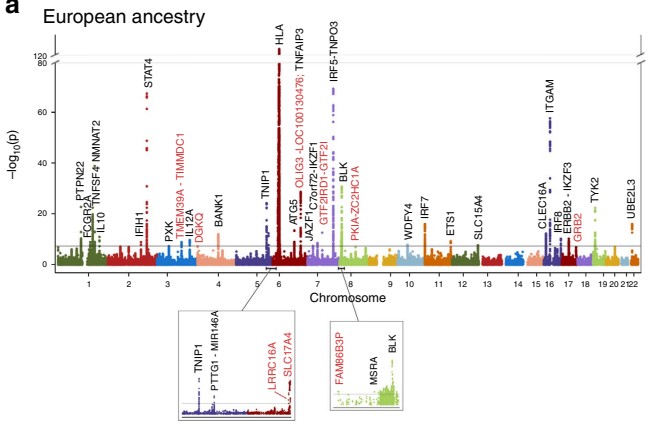

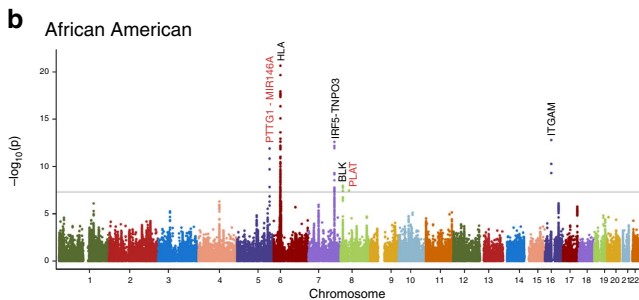

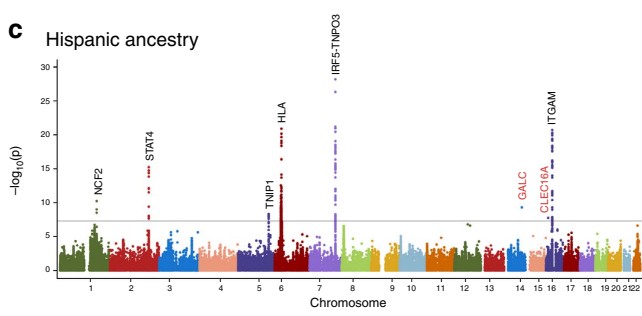

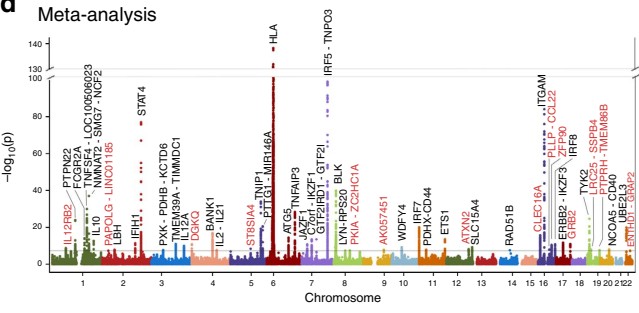

**Figure 1 | Genome-wide associations in SLE.** Manhattan plots for (**a**) European ancestry, (**b**) African American, (**c**) Hispanic ancestry, and the (**d**) meta-analysis. Tier 1 associations are labelled with novel regions highlighted in red. Genome-wide significance ($5 \times 10^{-8}$) is indicated on each plot.

22q12 (*C1QTNF6*). Only the 16p13 locus is associated in AA and EA.

**Chromosome X**. None of the 442 chromosome X SNPs, predominantly in Xp22 and Xq28, met Tier 1 or Tier 2 thresholds of significance. The strongest evidence of association was in females at Xq28 within *GAB3* (Supplementary Fig. 4; rs2664170;

EA: OR = 0.89, $P = 0.0009$; AA: OR = 0.90, $P = 0.13$; HA: OR = 0.90, $P = 0.33$; Meta $P = 1.23 \times 10^{-4}$).

**Two-way interactions among associated SNPs**. No SNP–SNP interactions met the Bonferroni threshold ($P = 1 \times 10^{-9}$) (see Methods).

**Human leukocyte antigen region**. SNP analyses within the HLA region provided strong evidence of association with SLE across groups (Fig. 2). These associations are complicated by the region's extended LD between SNPs and classical HLA alleles. Supplementary Data 3 and Supplementary Fig. 5 summarize the posterior probability distributions for the imputed four-digit HLA alleles in HLA-A, -B, -C, -DQA1, -DQB1, -DPB1 and -DRB1.

**HLA allele associations**. HLA allele associations for each ancestry and for multi-ancestral meta-analysis are shown in Supplementary Data 4. To disenable regional LD effects, ancestry-specific stepwise logistic modelling was used to identify the set of alleles with unique HLA contributions to SLE risk (that is, risk or 'protective' alleles associated even after adjusting for other SLE-associated HLA alleles) (Supplementary Data 5). To account for HLA alleles contributing even nominal effects, the models' entry and exit criteria were set to $P \leq 0.01$ (see Methods). The final models contained both risk and 'protective' alleles. In both the single-allele and multi-locus models, class II alleles exhibited the greatest association with SLE. The DR3 (DRB1*3:01-DQA1*05:01-DQB1*02:01) and DR15 (DRB1*15:01/ 03-DQA1*01:02-DQB1*06:01) haplotypes had the most significant class II risk alleles across populations.

**SNP associations after adjusting for HLA alleles**. Only two SNPs showed evidence of association with SLE (Supplementary Data 6) after adjusting for the HLA alleles identified in the stepwise modelling (Fig. 2). Specifically, for EA these SNPs are, rs1150755 (OR = 1.33, $P = 3.10 \times 10^{-8}$) within *TNXB* and rs9273448 (OR = 0.64, $P = 2.39 \times 10^{-8}$) within HLA-DQB1 (Supplementary Data 6 and Supplementary Fig. 6). These associations had comparable ORs in the AA and HA cohorts, except in HA for rs9273448. Transancestral meta-analysis showed stronger association at both loci (Supplementary Data 6 and Supplementary Fig. 6). Whether these residual associations reflect novel loci or imperfect imputation requires additional study.

**Compound risk allele heterozygosity**. In several autoimmune diseases, including lupus[11], having two different risk alleles (compound risk allele heterozygosity) generates greater disease risk than having two copies of the same risk allele[12,13]. In SLE, there are two primary risk haplotypes (DRB1*3:01-DQA1*05:01-DQB1*02:01 and DRB1*15-DQA1*01:02-DQB1*06:01), which are comprised of alleles in strong linkage disequilibrium. Thus, we selected DRB1*03:01 and DR*15 (DRB1*15:01 in EA & HA; DRB1*15:03 in AA) as tagging alleles to evaluate risk allele heterozygosity. Supplementary Data 7 summarizes the genotypic associations and contrasts the effects of risk allele homozygosity, heterozygosity, and compound heterozygosity. In both EA and AA, compound risk allele heterozygosity (DRB1*03:01/*15 provided greater risk than homozygosity for either individual risk allele (that is, DRB1*03:01/03:01; 15/15); these effects are consistent in direction but not significant in HA. Transancestral meta-analysis strongly supports that the risk for compound heterozygotes is greater than homozygotes for any individual allele ($P_{03:01} = 1.79 \times 10^{-10}$; $P_{15:01} = 4.65 \times 10^{-28}$). While there was not conclusive evidence of a statistical interaction for people having these two risk alleles in EA ($P = 0.07$), AA ($P = 0.06$),

**Table 2 | Novel ancestry-specific non-HLA associated regions.**

| Most significant SNP(s) | Chr. | Position (b37) | Gene region* | Region rank | Ref. allele | RAF case | RAF control | P value | OR (95% CI) | Regional stepwise P value | dbSNP function† |
|---|---|---|---|---|---|---|---|---|---|---|---|
| *EA Tier 1* | | | | | | | | | | | |
| rs1132200[d,P] | 3q13 | 119150836 | TMEM39A[1]-TIMMDC1 | 29 | A | 0.138 | 0.159 | $1.37 \times 10^{-7}$ | 0.83 (0.77-0.89) | | missense |
| rs1131265[d] | 3q13 | 119222456 | TMEM39A[1]-TIMMDC1 | 29 | C | 0.161 | 0.186 | $1.42 \times 10^{-9}$ | 0.81 (0.76-0.87) | $5.96 \times 10^{-9}$ | coding-synon |
| rs1534154 | 3q13 | 119311030 | TMEM39A[1]-TIMMDC1 | 29 | G | 0.177 | 0.165 | $2.60 \times 10^{-4}$ | 1.11 (1.05-1.18) | $1.35 \times 10^{-3}$ | |
| rs3733345 | 4p16 | 954247 | DGKQ | 33 | G | 0.444 | 0.466 | $5.84 \times 10^{-8}$ | 0.89 (0.85-0.93) | $5.84 \times 10^{-8}$ | untranslated-3 |
| rs4690229[i] | 4p16 | 970724 | DGKQ | 33 | T | 0.486 | 0.462 | $1.62 \times 10^{-8}$ | 1.13 (1.09-1.19) | | |
| rs10498722[d] | 6p22 | 25186512 | LRRC16A | 12 | A | 0.096 | 0.080 | $2.87 \times 10^{-10}$ | 1.30 (1.20-1.41) | $2.87 \times 10^{-10}$ | |
| rs35789010[i] | 6p22 | 25514179 | LRRC16A | 12 | A | 0.089 | 0.072 | $4.59 \times 10^{-19}$ | 1.46 (1.35-1.59) | | intron |
| rs4712969 | 6p22 | 25764192 | SLC17A4 | 8 | T | 0.119 | 0.093 | $1.83 \times 10^{-22}$ | 1.42 (1.32-1.52) | $1.83 \times 10^{-22}$ | intron |
| rs36014129[i] | 6p22 | 25884845 | SLC17A4 | 8 | A | 0.101 | 0.079 | $1.21 \times 10^{-24}$ | 1.54 (1.39-1.62) | | |
| rs2327832 | 6q23 | 137973068 | OLIG3-LOC100130476[‡] | 6 | C | 0.239 | 0.212 | $1.76 \times 10^{-13}$ | 1.22 (1.15-1.28) | $2.38 \times 10^{-8}$ | |
| rs17779870 | 6q23 | 138156425 | OLIG3-LOC100130476[‡] | 6 | C | 0.132 | 0.154 | $5.35 \times 10^{-7}$ | 0.85 (0.80-0.91) | $1.80 \times 10^{-5}$ | intron |
| rs5029939 | 6q23 | 138195723 | TNFAIP3[‡] | 6 | C | 0.061 | 0.033 | $2.39 \times 10^{-29}$ | 1.81 (1.63-2.01) | $7.21 \times 10^{-22}$ | intron |
| rs2230926[P] | 6q23 | 138196066 | TNFAIP3[‡] | 6 | C | 0.061 | 0.033 | $2.79 \times 10^{-29}$ | 1.81 (1.63-2.01) | | missense |
| rs77000060[i] | 6q23 | 138237989 | TNFAIP3[‡] | 6 | T | 0.055 | 0.030 | $1.84 \times 10^{-29}$ | 1.89 (1.69-2.11) | | |
| rs73137125 | 7q11 | 74018950 | GTF2IRD1-GTF2I[1] | 15 | G | 0.219 | 0.230 | $3.27 \times 10^{-3}$ | 0.92 (0.88-0.97) | $1.08 \times 10^{-5}$ | |
| rs73366469 | 7q11 | 74033600 | GTF2IRD1-GTF2I[1] | 15 | C | 0.126 | 0.098 | $2.68 \times 10^{-13}$ | 1.29 (1.21-1.38) | $1.11 \times 10^{-15}$ | |
| rs2955587 | 8p23 | 8098079 | FAM86B3P | 25 | C | 0.468 | 0.442 | $7.91 \times 10^{-10}$ | 1.15 (1.10-1.20) | $7.91 \times 10^{-10}$ | intron |
| rs2980512[i] | 8p23 | 8140901 | FAM86B3P | 25 | C | 0.497 | 0.467 | $3.54 \times 10^{-10}$ | 1.15 (1.10-1.20) | | |
| rs1966115 | 8q21 | 79556891 | PKIA-ZC2HC1A | 23 | A | 0.296 | 0.256 | $1.43 \times 10^{-7}$ | 1.14 (1.09-1.20) | $4.11 \times 10^{-11}$ | |
| rs12114284 | 8q21 | 79558441 | PKIA-ZC2HC1A | 23 | A | 0.291 | 0.256 | $2.75 \times 10^{-7}$ | 1.11 (1.06-1.17) | $1.93 \times 10^{-10}$ | |
| rs930297 | 17q25 | 73404537 | GRB2 | 38 | G | 0.106 | 0.116 | $1.43 \times 10^{-7}$ | 0.83 (0.77-0.89) | $4.91 \times 10^{-8}$ | |
| rs1463485[d] | 17q25 | 73851791 | GRB2 | 38 | G | 0.223 | 0.197 | $7.34 \times 10^{-5}$ | 1.14 (1.07-1.21) | $2.27 \times 10^{-5}$ | near-gene-5 |
| *EA Tier 2* | | | | | | | | | | | |
| rs11590283[i] | 1p36 | 1245368 | CPSF3L | 42 | G | 0.188 | 0.211 | $1.36 \times 10^{-7}$ | 0.86 (0.82-0.91) | | intron |
| rs12142199 | 1p36 | 1249187 | CPSF3L | 42 | C | 0.189 | 0.212 | $1.89 \times 10^{-7}$ | 0.87 (0.82-0.91) | $1.89 \times 10^{-7}$ | coding-synon |
| rs6662618[d] | 1p22 | 92935411 | GFI1-EVI5 | 50 | T | 0.182 | 0.157 | $1.54 \times 10^{-6}$ | 1.18 (1.10-1.26) | $3.81 \times 10^{-3}$ | intron |
| rs12738833 | 1p22 | 93119118 | GFI1-EVI5 | 50 | G | 0.273 | 0.251 | $7.66 \times 10^{-6}$ | 1.12 (1.06-1.17) | $8.34 \times 10^{-3}$ | intron |
| rs11578098 | 1p22 | 93119410 | GFI1-EVI5 | 50 | A | 0.275 | 0.250 | $5.42 \times 10^{-7}$ | 1.13 (1.08-1.19) | $4.01 \times 10^{-7}$ | intron |
| rs41264285[i] | 1q22 | 155033918 | ADAM15-EFNA1 | 54 | T | 0.226 | 0.211 | $8.29 \times 10^{-7}$ | 1.14 (1.08-1.20) | | coding-synon, intron, missense |
| rs45444697 | 1q22 | 155034632 | ADAM15-EFNA1 | 54 | G | 0.225 | 0.210 | $1.39 \times 10^{-6}$ | 1.14 (1.08-1.20) | $1.83 \times 10^{-5}$ | intron, near-gene-5 |
| rs4971066[d] | 1q22 | 155105882 | ADAM15-EFNA1 | 54 | G | 0.147 | 0.168 | $4.02 \times 10^{-5}$ | 0.87 (0.81-0.93) | $4.47 \times 10^{-4}$ | intron |
| rs6756736[r] | 2p21 | 43558743 | THADA | 45 | T | 0.211 | 0.218 | $5.67 \times 10^{-3}$ | 1.22 (1.06-1.41) | $8.30 \times 10^{-6}$ | intron |
| rs6705304 | 2p21 | 43596746 | THADA | 45 | C | 0.083 | 0.099 | $9.06 \times 10^{-5}$ | 0.86 (0.80-0.93) | $1.75 \times 10^{-7}$ | intron |
| rs62149377 | 2p16 | 60986576 | PAPOLG | 39 | G | 0.302 | 0.275 | $9.22 \times 10^{-6}$ | 1.14 (1.09-1.19) | $1.45 \times 10^{-5}$ | intron |
| rs115291397[d] | 2p16 | 61060043 | PAPOLG | 39 | C | 0.007 | 0.012 | $4.55 \times 10^{-5}$ | 0.61 (0.48-0.77) | $1.56 \times 10^{-4}$ | |
| rs2600669 | 2p15 | 61401296 | PAPOLG | 39 | T | 0.366 | 0.387 | $1.10 \times 10^{-5}$ | 0.90 (0.86-0.95) | $5.70 \times 10^{-4}$ | |
| rs115268109[d] | 2p15 | 61833802 | LOC100132037-FLJ13305 | 47 | C | 0.055 | 0.045 | $2.38 \times 10^{-7}$ | 1.31 (1.18-1.45) | $2.38 \times 10^{-7}$ | |
| rs11681718 | 2q12 | 103051144 | IL18RAP | 40 | C | 0.252 | 0.287 | $1.18 \times 10^{-7}$ | 0.88 (0.83-0.92) | $1.18 \times 10^{-7}$ | intron |
| rs2460382[d] | 2q21 | 135014116 | MGAT5[§] | 52 | C | 0.229 | 0.232 | $2.82 \times 10^{-3}$ | 0.91 (0.85-0.97) | $1.22 \times 10^{-3}$ | intron |
| rs10496726 | 2q21 | 135045250 | MGAT5[§] | 52 | C | 0.094 | 0.094 | $3.87 \times 10^{-5}$ | 0.85 (0.79-0.92) | $1.86 \times 10^{-5}$ | intron |
| rs11887156[i] | 2q21 | 135066476 | MGAT5[§] | 52 | C | 0.113 | 0.115 | $5.29 \times 10^{-5}$ | 0.84 (0.78-0.90) | | intron |
| rs2196171[i] | 2q33 | 198889807 | PLCL1 | 51 | T | 0.465 | 0.500 | $4.65 \times 10^{-7}$ | 0.89 (0.85-0.93) | | intron |
| rs6738825 | 2q33 | 198896895 | PLCL1 | 51 | T | 0.460 | 0.494 | $1.27 \times 10^{-6}$ | 0.90 (0.86-0.94) | $1.27 \times 10^{-6}$ | intron |
| rs4921317[d] | 5q33 | 158538277 | LOC285627 | 49 | C | 0.480 | 0.470 | $1.46 \times 10^{-4}$ | 1.17 (1.09-1.25) | $1.94 \times 10^{-5}$ | |
| rs6869688 | 5q33 | 158883027 | LOC285627 | 49 | C | 0.467 | 0.493 | $3.38 \times 10^{-7}$ | 0.89 (0.85-0.93) | $3.97 \times 10^{-7}$ | intron |
| rs7720046[i] | 5q33 | 158884535 | LOC285627 | 49 | G | 0.467 | 0.493 | $2.92 \times 10^{-7}$ | 0.89 (0.85-0.93) | | intron |
| rs71567468[i] | 6p21 | 34816070 | DEF6-PPARD[1] | 56 | T | 0.057 | 0.042 | $8.70 \times 10^{-7}$ | 1.29 (1.17-1.43) | | intron |
| rs6920432[d] | 6p21 | 35298662 | DEF6-PPARD[1] | 56 | G | 0.100 | 0.080 | $7.40 \times 10^{-4}$ | 1.15 (1.06-1.25) | $7.40 \times 10^{-4}$ | |
| rs1039917 | 8p23 | 8718850 | MFHAS1 | 43 | A | 0.397 | 0.375 | $1.48 \times 10^{-7}$ | 1.13 (1.08-1.18) | $1.48 \times 10^{-7}$ | intron |
| rs12156002 | 8q24 | 129190544 | PVT1-BC009730 | 48 | A | 0.194 | 0.221 | $2.87 \times 10^{-7}$ | 0.87 (0.82-0.92) | $2.44 \times 10^{-7}$ | |
| rs6651252[d] | 8q24 | 129567181 | PVT1-BC009730 | 48 | C | 0.119 | 0.131 | $8.80 \times 10^{-6}$ | 0.85 (0.79-0.91) | $8.22 \times 10^{-6}$ | |
| rs11788118 | 9q22 | 102337331 | AK057451 | 57 | A | 0.205 | 0.224 | $1.07 \times 10^{-6}$ | 0.88 (0.83-0.92) | $1.07 \times 10^{-6}$ | |
| rs10819689[i] | 9q22 | 102400263 | AK057451 | 57 | T | 0.202 | 0.221 | $9.26 \times 10^{-7}$ | 0.87 (0.83-0.92) | | |
| rs12722558 | 10p15 | 6070274 | IL2RA | 46 | A | 0.124 | 0.114 | $2.69 \times 10^{-3}$ | 1.11 (1.04-1.18) | $9.40 \times 10^{-5}$ | intron |
| rs10905718 | 10p15 | 6114856 | IL2RA | 46 | G | 0.319 | 0.308 | $3.99 \times 10^{-6}$ | 1.12 (1.07-1.17) | $1.86 \times 10^{-7}$ | |
| rs112123005 | 10p15 | 6472492 | IL2RA | 46 | C | 0.024 | 0.020 | $1.64 \times 10^{-4}$ | 1.33 (1.14-1.53) | $2.75 \times 10^{-4}$ | intron |
| rs113304138[d] | 10p15 | 6564277 | IL2RA | 46 | G | 0.008 | 0.013 | $2.03 \times 10^{-4}$ | 0.65 (0.51-0.81) | $2.78 \times 10^{-4}$ | intron |
| rs223881 | 16q13 | 57386566 | PLLP-CCL22 | 44 | T | 0.263 | 0.236 | $3.19 \times 10^{-7}$ | 1.14 (1.08-1.20) | $3.19 \times 10^{-7}$ | |
| rs223883[i] | 16q13 | 57388730 | PLLP-CCL22 | 44 | G | 0.250 | 0.224 | $1.64 \times 10^{-7}$ | 1.15 (1.09-1.21) | | |
| rs1170436[d] | 16q22 | 68607486 | ZFP90[1] | 55 | A | 0.248 | 0.221 | $8.50 \times 10^{-7}$ | 1.17 (1.10-1.25) | $8.50 \times 10^{-7}$ | |
| rs11673460[d] | 19p13 | 18191621 | LRRC25-SSBP4 | 53 | T | 0.054 | 0.060 | $3.41 \times 10^{-3}$ | 0.86 (0.78-0.95) | $5.84 \times 10^{-4}$ | intron |
| rs425648 | 19p13 | 18202112 | LRRC25-SSBP4 | 53 | A | 0.177 | 0.196 | $2.59 \times 10^{-4}$ | 0.90 (0.85-0.95) | $2.18 \times 10^{-4}$ | |
| rs12971295[i] | 19p13 | 18517331 | LRRC25-SSBP4 | 53 | A | 0.258 | 0.288 | $6.67 \times 10^{-7}$ | 0.88 (0.84-0.93) | | |
| rs13344313 | 19p13 | 18517767 | LRRC25-SSBP4 | 53 | A | 0.259 | 0.290 | $7.03 \times 10^{-7}$ | 0.88 (0.84-0.93) | $1.36 \times 10^{-6}$ | |
| *AA Tier 1* | | | | | | | | | | | |
| rs2431697 | 5q33 | 159879978 | PTTG1-MIR146A[1] | 3 | C | 0.398 | 0.467 | $1.27 \times 10^{-12}$ | 0.76 (0.70-0.82) | $1.27 \times 10^{-12}$ | |
| rs1804182[d] | 8p11 | 42033519 | PLAT | 5 | A | 0.042 | 0.022 | $3.48 \times 10^{-8}$ | 1.94 (1.53-2.45) | $3.48 \times 10^{-8}$ | nonsense |
| *AA Tier 2* | | | | | | | | | | | |
| rs34840245 | 6p21 | 34812701 | UHRF1BP1-DEF6[1] | 7 | G | 0.273 | 0.237 | $2.49 \times 10^{-5}$ | 1.21 (1.11-1.32) | $2.02 \times 10^{-3}$ | intron |
| rs1194[d] | 6p21 | 35263555 | UHRF1BP1-DEF6[1] | 7 | A | 0.377 | 0.334 | $5.04 \times 10^{-7}$ | 1.32 (1.19-1.48) | $3.69 \times 10^{-5}$ | |
| rs1170436[d] | 16q22 | 68607486 | ZFP90[1] | 9 | A | 0.281 | 0.244 | $7.93 \times 10^{-7}$ | 1.31 (1.18-1.46) | $7.93 \times 10^{-7}$ | |
| *HA Tier 1* | | | | | | | | | | | |
| rs11845506[d] | 14q31 | 88383035 | GALC | 5 | A | 0.005 | 0.024 | $5.00 \times 10^{-10}$ | 0.20 (0.12-0.33) | $5.00 \times 10^{-10}$ | |
| rs8054198[d] | 16p13 | 11038360 | CLEC16A[1] | 7 | T | 0.011 | 0.031 | $1.79 \times 10^{-8}$ | 0.36 (0.25-0.51) | $1.26 \times 10^{-8}$ | untranslated-5 |
| rs12448240[d] | 16p13 | 11187218 | CLEC16A[1] | 7 | G | 0.018 | 0.011 | $3.76 \times 10^{-4}$ | 2.06 (1.38-3.06) | $2.07 \times 10^{-5}$ | intron |
| rs12925552 | 16p13 | 11332805 | CLEC16A[1] | 7 | C | 0.220 | 0.275 | $1.19 \times 10^{-3}$ | 0.84 (0.75-0.93) | $4.77 \times 10^{-5}$ | |

**Table 2 (Continued).**

| Most significant SNP(s) | Chr. | Position (b37) | Gene region* | Region rank | Ref. allele | RAF case | RAF control | P value | OR (95% CI) | Regional stepwise P value | dbSNP function† |
|---|---|---|---|---|---|---|---|---|---|---|---|
| *HA Tier 2* | | | | | | | | | | | |
| rs73846279[i] | 3p11 | 89891345 | EPHA3-PROS1 | 15 | T | 0.088 | 0.061 | $4.82 \times 10^{-7}$ | 1.57 (1.32–1.88) | | |
| rs7653338[d] | 3p11 | 89938088 | EPHA3-PROS1 | 15 | A | 0.087 | 0.060 | $1.64 \times 10^{-6}$ | 1.58 (1.31–1.91) | $1.64 \times 10^{-6}$ | |
| rs9394274 | 6p21 | 35114911 | TCP11-SCUBE3[1] | 8 | A | 0.285 | 0.243 | $6.53 \times 10^{-8}$ | 1.34 (1.20–1.49) | $6.53 \times 10^{-8}$ | |
| rs12199481[d] | 6q25 | 159381492 | RSPH3 | 16 | C | 0.430 | 0.396 | $5.72 \times 10^{-4}$ | 0.78 (0.67–0.90) | $2.92 \times 10^{-5}$ | |
| rs2092540[d] | 6q25 | 159416444 | RSPH3 | 16 | T | 0.145 | 0.198 | $8.92 \times 10^{-6}$ | 0.73 (0.63–0.84) | $6.23 \times 10^{-7}$ | intron |
| rs2041862[d] | 12q15 | 68461697 | DYRK2-IFNG | 12 | A | 0.100 | 0.150 | $1.59 \times 10^{-7}$ | 0.66 (0.57–0.77) | $1.59 \times 10^{-7}$ | |
| rs17005500[d] | 12q21 | 79738884 | SYT1 | 13 | C | 0.048 | 0.071 | $2.43 \times 10^{-7}$ | 0.58 (0.47–0.71) | $2.43 \times 10^{-7}$ | intron |
| rs2550333 | 16q21 | 58267472 | CSNK2A2-CCDC113 | 10 | G | 0.374 | 0.292 | $9.52 \times 10^{-7}$ | 1.28 (1.16–1.41) | $9.52 \times 10^{-7}$ | |
| rs2731763[i] | 16q21 | 58280078 | CSNK2A2-CCDC113 | 10 | G | 0.364 | 0.279 | $1.03 \times 10^{-7}$ | 1.31 (1.19–1.45) | | |
| rs229533 | 22q12 | 37587111 | C1QTNF6 | 14 | C | 0.488 | 0.437 | $4.61 \times 10^{-6}$ | 1.24 (1.13–1.35) | $1.41 \times 10^{-2}$ | |
| rs229541 | 22q12 | 37591318 | C1QTNF6 | 14 | T | 0.486 | 0.427 | $2.46 \times 10^{-7}$ | 1.27 (1.16–1.39) | $1.21 \times 10^{-3}$ | |

Novel regions have not previously been identified by SNP associations with P values $<5 \times 10^{-8}$ and are highlighted in grey. Regions that are the first observed associations in a particular ancestry are indicated with a superscript[1] in the gene region.
[i]: Imputed SNP.
[d] or [r]: Dominant, or recessive model; if not noted, additive model was used.
[P]: Published association—this SNP has been identified as causal or as the most significant SNP in gene region.
*Named by the genes bounding the region of association, unless literature strongly implicated a specific gene.
†dbSNP's predicted functional effect.
‡The OLIG3-LOC100130476 region reaches Tier 1 significance even after adjusting for the TNFAIP3 signal, an established SLE region.
§We validated that the MGAT5 region is distinct from the Lactase gene (LCT) on Chromosome 2, by adjusting for the top hit in the LCT gene (rs55634455[i], $P = 4.77 \times 10^{-4}$, OR = 0.87). After this adjustment, the top SNPs were still significant and minimally affected by the adjustment (rs2460382[d]: $P = 5.17 \times 10^{-3}$, OR = 0.91; rs10496726: $P = 6.65 \times 10^{-5}$, OR = 0.86).

or HA ($P = 0.50$), the lack-of-fit test supported the dominance model of risk (departure from additivity; see Methods) for an individual DR3 (EA $P = 7.90 \times 10^{-109}$; AA $P = 0.06$; HA $P = 5.14 \times 10^{-10}$) and DR15 (EA $P = 5.79 \times 10^{-26}$; AA $P = 3.99 \times 10^{-13}$; HA $P = 3.25 \times 10^{-11}$) SLE risk alleles.

**HLA clustering by amino acid.** HLA alleles with high sequence similarity, but contrasting ORs, suggest the potential presence of key amino acids influencing disease risk. As expected, clustering amino acid sequences resulted in most two-digit allele subtypes residing within the same clusters (Fig. 3 and Supplementary Fig. 7). When evaluating SLE associations of the three ancestries across these sequence clusters, several noteworthy patterns emerged.

The two primary DRB1 risk alleles, DR3 and DR15 clustered separately, suggesting comparative amino acid dissimilarity. Notably, the closest-clustered neighbours to each risk allele conferred non-risk in these three ancestries. Multi-sequence alignment distinguished the unique or less common amino acids among risk alleles (Supplementary Figs 8–10). Unique to risk alleles DRB1*15:01 and *15:03 were the amino acids Ser-1 (signal peptide), Phe47 and Ala71. Three-dimensional modelling of DRB1 (Supplementary Fig. 8b,c) reveals that these differences mostly reside within the peptide-binding pocket, creating a space of non-polar (hydrophobic) residues, unlike the polar-residue (hydrophilic) space of Tyr47 and Arg71 or Glu71 provided by non-risk alleles within this cluster (Supplementary Fig. 9). Residue 71, among the most variable residues in DRB1 (ref. 14), has been implicated in other diseases[15]. Among non-risk alleles with at least 95% identity to DRB1*03:01, the only amino acid unique to this risk allele was Tyr26 (Supplementary Fig. 10). DRB1*03:01 amino acids shared by less than half of the non-risk alleles in this cluster are highlighted in Supplementary Fig. 10 and are concentrated between positions 70–77, spanning the designated 'Shared Epitope' region[16,17].

One predominant DQA-DQB1 pair of SLE risk alleles exists per evolutionary DQ-sublineage (Fig. 3b,c)[18]. In the DQ2/3/4 sublineage, DQA1*05:01 confers risk across the three cohorts and its heterodimer counterpart, DQB1*02:01, confers risk in EA and HA, but not significantly in AA. Within the DQ5/6 sublineage, both DQA1*01:02 and DQB1*06:02 yield SLE risk across all three cohorts. Comparison of DQA1*01:02 to its closest-related alleles (Supplementary Fig. 11) reveals that DQA1*01:02 (DR15) uniquely encodes a Met207 versus Val207. DQA1*05:01

encodes a polar Thr13 compared to the non-polar Ala13 found in DQA1*05:05 (DR3) and DQA1*05:03 (Supplementary Fig. 12). Identification of specific risk residues was less distinct for the DQB1 risk alleles.

**Gender-HLA and genome-wide SNP-HLA interaction.** There was no evidence that the risk of SLE differed by gender at any HLA alleles or of a significant SNP-by-HLA allele interaction anywhere across the genome ($P_{FDR} > 0.05$).

**Transancestral mapping and top meta-analysis regions.** The three-ancestry meta-analysis identified additional SLE-associated regions and was particularly informative for 22 regions, including 11 novel regions, 3 published regions that now meet genome-significance, a complex multigenic region identified by adjusting for HLA alleles and 7 well-established regions more sharply localized by transancestral mapping or novel to these ancestries (Tables 3 and 4; Supplementary Figs 13–15). Supplementary Data 8 and Supplementary Fig. 16 show additional regions that only met genome-wide significance in the meta-analysis. Supplementary Data 9 lists any region with meta-analysis $P_{FDR} < 0.001$.

On 1p31, rs3828069 is within an intron of *IL12RB2* (OR = 0.85, $P = 1.77 \times 10^{-9}$) and has evidence of association in all three ancestries. Although *IL12RB2* is implicated in multiple autoimmune diseases[19,20], this specific SNP association with SLE is novel. The 2p16 region exhibited a novel SLE association at rs1432296 (OR = 1.18, $P = 1.34 \times 10^{-8}$) near *PAPOLG-LINC01185*, which includes *REL*. A linkage region at 4p16 (ref. 21) contained a strong novel association for rs3733345 (OR = 0.89, $P = 1.83 \times 10^{-11}$); EA dominated the association, but with significant support from HA and AA. On 8q21, rs4739134 is near *PKIA-ZC2HC1A* (OR = 1.12, $P = 3.47 \times 10^{-8}$) and the AA helped localize the association. The region about 16q13 (*PLLP-CCL22*) exhibited modest association in individual ancestries, but reached genome-wide significance for rs223889 (OR = 1.21, $P = 1.08 \times 10^{-8}$) in the meta-analysis. Similarly, rs137956 (OR = 0.88, $P = 5.0 \times 10^{-8}$) on 22q13 between *ENTHD1* and *GRAP2* was supported across all three ancestries. We bioinformatically explore three additional novel regions.

The meta-analysis about 16q22 (rs1749792; OR = 1.14, $P = 3.66 \times 10^{-11}$) near *ZFP90* had strong support from both EA and AA, with AA samples localizing the association

(Supplementary Fig. 13l). While previously identified in a Chinese cohort, this is the first significant association within EA and AA[8]. Within this region, 27 additional SNPs had a meta-analysis $P$ value within one order of magnitude of the maximum association, rs1749792. These 28 SNPs span an interval of 44.6 kb, narrowed from the 100 kb associated region in EA. RegulomeDB[22] and HaploReg4.1 (ref. 23) identified 4 of these SNPs with a RegulomeDB score of 1f and 1 with a RegulomeDB score of 2f, indicating they were eQTLs and transcription factor binding sites. HaploReg4.1 showed these five SNPs were enhancers and promotor histone marks in multiple tissues. Interestingly, one of these five, rs1170445, is in high LD with rs1749792 ($R^2_{EA} = 0.99$, $R^2_{AA} = 0.84$, $R^2_{HA} = 0.99$). Here, the G allele is the risk allele and creates a CpG site in the promoter region. In GTEx, the G allele corresponds to lowest gene expression. Hence, when methylated, this variant should result in decreased gene expression of $ZFP90$. The rs1170445-$ZFP90$ expression association was reported in GTEx for whole blood ($P = 1 \times 10^{-47}$) and several other tissues (that is, spleen, skeletal muscle, brain cortex, lung, testis and EBV-transformed lymphocytes). Huang et al.[24] found expression of $ZFP90$ in Jurkat T cells led to decreased expression of IL2 and interferon. Furthermore, they found that $ZFP90$ protein binds to IL2 and interferon gamma promoters.

$SLC15A4$ was associated with SLE in the EA cohort and localized by the AA signal in the meta-analysis. The top EA signal was supported by a 43.7 kb region of SLE-associated SNPs exhibiting $P$ values within one order of magnitude of the top signal. The meta-analysis narrowed the region of association to four SNPs, spanning 9.5 kb around rs1059312 (Supplementary Fig. 15j). rs1059312 is an eQTL for $SLC15A4$ and three supporting SNPs (rs2291349, rs4760593 and rs11059916) altered CpG sites. The region has been previously reported in Asian populations[25,26]; but this is the first instance of genome-wide significance in EA ($P < 5 \times 10^{-8}$)[26].

On 17q25 near $GRB2$, rs8072449 (OR = 0.84, $P = 1.19 \times 10^{-11}$) had modest support in each ancestry, but met genome-wide significance and better localization in the meta-analysis. rs8072449 is an eQTL for $GRB2$ (Supplementary Fig. 13m). There were eight additional SNPs with a meta-analysis $P$ value within one order of magnitude of the maximum association, and the transancestral analysis reduced the interval of association from 93 to 82 kb. The best RegulomeDB scores for these 9 SNPs was 1f for rs7219, reflecting rs7219 as a known cis-eQTL ($NUP85$, $MIF4GD$, $MRPS7$), a transcription binding site and within a DNase peak; in total 7 of the 9 SNPs were reported in transcription binding sites. Interestingly, the top associated SNP, rs8072449, breaks a CpG site and 6 others either end or begin a CpG site. Hence, 7 of the 9 top associated SNPs make or break a CpG site and several are transcription binding sites. Of the 147,111 Immunochip SNPs that passed quality control analyses, only 30% begin or end a CpG site. Although this is a novel SLE association, GRB2 reportedly regulates SHP2 activity[27,28], a potential contributor to SLE pathogenesis[29].

A few novel regions, sparsely mapped on the Immunochip, reached genome-wide significance in the meta-analysis and merit further fine-mapping efforts. These include rs6886392 on 5q21 (OR = 1.13, $P = 4.08 \times 10^{-9}$), rs11788118 on 9q22 (OR = 0.88, $P = 1.53 \times 10^{-8}$) and rs13344313 on 19p13 (OR = 0.90, $P = 1.07 \times 10^{-8}$).

Additional loci not previously reported as having genome-wide significance for SLE in these ancestries now do so in the meta-analysis (Table 4). On 4q27, rs11724582 (OR = 0.88, $P = 1.71 \times 10^{-8}$) is near $IL21$, a known SLE risk locus[30,31]. $IL21$ is up-regulated by oestrogen and is produced by T follicular helper cells which stimulates B-cells to differentiate into autoantibody-secreting cells; however, there was no evidence of a SNP-by-gender interaction in any ancestry ($P > 0.40$). The SNP rs2431098 (OR = 1.19, $P = 3.29 \times 10^{-21}$) at 5q33 between $PTTG1$ and $MIR146A$ has an $r^2 = 0.52$ with rs2431697, a SNP correlated with down-regulation of $MIR146A$[32].

The 6p21 region is potentially confounded with nearby HLA associations. The advantages of using multiple ancestries in this study are exemplified by modelling of SNPs in the 6p21 region where three separate ancestry-specific signals were identified after adjusting for HLA alleles. The results show associations at previously reported $UHRF1BP1$ and two novel loci within the $SCUBE3$-$DEF6$ region (Fig. 2 and Supplementary Fig. 13e,f).

The transancestral meta-analyses of several previously established SLE associations provided important localization, and increased the number of independent signals or novel transancestral effects. These included: 1q25 ($TNFSF4$-$LOC100506023$), 1q25 ($NMNAT2$-$SMG7$-$NCF2$), 7q32 ($IRF5$-$TNPO3$), 8q12 ($LYN$-$RPS20$), 11p13 ($PDHX$-$CD44$) and 20q13 ($NCOA5$-$CD40$) (Table 4, Supplementary Fig. 15).

**Admixture and population frequencies of SLE-associated SNPs.** Clustering risk allele frequencies for Tier 1 and 2 SNPs in cases across EA, AA, and HA yielded three groups of SNPs: comparable allele frequencies in all three ancestries (75 SNPS), increased frequency in AA cases (40 SNPs), and reduced frequency in AA cases (66 SNPs) (Fig. 4); the latter two clusters show increased and decreased AA-ancestral contribution, respectively. Higher frequency risk alleles tend to exhibit comparable frequencies across ancestries; the rarest alleles were largely grouped in the reduced AA-ancestral cluster. When comparing admixture averages for risk alleles, AA exhibited the highest deviations from mean admixture estimates and EA, the lowest (Fig. 4; Supplementary Data 10). Deviations from average admixture in risk alleles were significantly weighted to higher proportions of CEU versus YRI in AA ($P = 8.36 \times 10^{-12}$) and HA ($P = 2.44 \times 10^{-4}$) (Supplementary Data 11), further suggesting increased European ancestry for risk alleles. When aligned to allele frequency information, highest CEU proportion deviations in AA and HA resided in the decreased-AA cluster, while the YRI proportion deviations resided in the increased-AA cluster. Thus, SLE risk alleles with a low frequency in AA are correlated with European admixture. Of the 181 Tier 1 and 2 SNPs, only in two regions were the top associated SNP (rs1804182 AA Tier 1 and rs11845506 HA Tier 2) nearly monomorphic (frequency < 0.003) in the other ancestral cohorts. This suggests that most of the ancestry-specific SNP associations were not driven by the presence of monomorphic alleles in the non-discovery cohorts. These allele patterns are further illustrated in Fig. 4.

**Genetic load and SLE risk.** To explore effects of the number of risk polymorphisms on SLE risk, we computed the genetic risk allele load (unweighted and β-weighted (β = log(OR)), see Methods). Here, a set of ORs that contrasted the lowest 10% of the risk-allele count distribution with a sliding window of 20 unweighted, or 4 weighted, counts was computed; these logistic models adjusted for admixture. The pattern of the sliding window ORs was different across ancestries (Fig. 5 and Table 5). Specifically, in 2,000 EA cases and 2,000 EA controls that were independent from the discovery set, a strong and nonlinear effect emerged, with $OR_{unweighted} > 30$ and $OR_{weighted} > 100$ for the highest load groups. In fact, there was a nonlinear trend in the log(OR) (that is, β parameter denoting slope) with a greater than additive effect at the highest quarter of the genetic load range (Supplementary Fig. 17); this pattern suggests that the effect of at

**Table 3 | Novel non-HLA associated regions identified by transancestral meta-analysis.**

| SNP | Chr. | Position (b37) | Gene region* | Ref. allele | Ancestry | RAF case | RAF control | P value | OR (95% CI) | dbSNP function[†] |
|---|---|---|---|---|---|---|---|---|---|---|
| *Tier 1 Meta-Analysis* | | | | | | | | | | |
| rs3828069 | 1p31 | 67839573 | IL12RB2 | G | Meta | – | – | $1.77 \times 10^{-9}$ | 0.85 (0.79–0.90) | intron |
| | | | | | EA | 0.164 | 0.182 | $3.37 \times 10^{-6}$ | 0.87 (0.82–0.92) | |
| | | | | | AA | 0.042 | 0.050 | $3.13 \times 10^{-2}$ | 0.82 (0.68–0.98) | |
| | | | | | HA | 0.200 | 0.225 | $6.43 \times 10^{-4}$ | 0.82 (0.74–0.92) | |
| rs1432296 | 2p16 | 61068167 | PAPOLG-LINC01185 | A | Meta | – | – | $1.34 \times 10^{-8}$ | 1.18 (1.10–1.26) | |
| | | | | | EA | 0.165 | 0.147 | $5.31 \times 10^{-7}$ | 1.17 (1.10–1.24) | |
| | | | | | AA | 0.049 | 0.042 | $6.41 \times 10^{-2}$ | 1.19 (0.99–1.42) | |
| | | | | | HA | 0.083 | 0.083 | $3.80 \times 10^{-2}$ | 1.19 (1.01–1.41) | |
| rs3733345 | 4p16 | 954247 | DGKQ | G | Meta | – | – | $1.83 \times 10^{-11}$ | 0.89 (0.85–0.92) | untranslated-3 |
| | | | | | EA | 0.444 | 0.466 | $5.84 \times 10^{-8}$ | 0.89 (0.85–0.93) | |
| | | | | | AA | 0.455 | 0.482 | $3.56 \times 10^{-3}$ | 0.89 (0.83–0.96) | |
| | | | | | HA | 0.358 | 0.403 | $7.03 \times 10^{-3}$ | 0.88 (0.80–0.97) | |
| rs6886392 | 5q21 | 100135865 | ST8SIA4 | C | Meta | – | – | $4.08 \times 10^{-9}$ | 1.13 (1.08–1.18) | |
| | | | | | EA | 0.314 | 0.297 | $5.16 \times 10^{-6}$ | 1.12 (1.06–1.17) | |
| | | | | | AA | 0.257 | 0.235 | $8.53 \times 10^{-3}$ | 1.12 (1.03–1.23) | |
| | | | | | HA | 0.236 | 0.224 | $7.44 \times 10^{-3}$ | 1.16 (1.04–1.29) | |
| rs34840245 | 6p21 | 34812701 | UHRF1BP1-DEF6[1] | G | Meta | – | – | $2.37 \times 10^{-11}$ | 1.20 (1.14–1.27) | intron |
| | | | | | EA | 0.130 | 0.106 | $4.03 \times 10^{-6}$ | 1.18 (1.10–1.26) | |
| | | | | | AA | 0.273 | 0.237 | $9.06 \times 10^{-5}$ | 1.20 (1.09–1.31) | |
| | | | | | HA | 0.122 | 0.101 | $1.47 \times 10^{-3}$ | 1.27 (1.10–1.48) | |
| rs4739134 | 8q21 | 79556148 | PKIA-ZC2HC1A | T | Meta | – | – | $3.47 \times 10^{-8}$ | 1.12 (1.07–1.17) | |
| | | | | | EA | 0.290 | 0.255 | $2.80 \times 10^{-5}$ | 1.11 (1.06–1.17) | |
| | | | | | AA | 0.408 | 0.377 | $1.30 \times 10^{-3}$ | 1.14 (1.05–1.23) | |
| | | | | | HA | 0.195 | 0.197 | $7.00 \times 10^{-2}$ | 1.11 (0.99–1.25) | |
| rs11788118 | 9q22 | 102337331 | AK057451 | A | Meta | – | – | $1.53 \times 10^{-8}$ | 0.88 (0.84–0.92) | |
| | | | | | EA | 0.205 | 0.224 | $1.07 \times 10^{-6}$ | 0.88 (0.83–0.92) | |
| | | | | | AA | 0.160 | 0.167 | $4.12 \times 10^{-1}$ | 0.96 (0.87–1.06) | |
| | | | | | HA | 0.146 | 0.187 | $4.23 \times 10^{-4}$ | 0.80 (0.71–0.91) | |
| rs653178 | 12q24 | 112007756 | ATXN2 | C | Meta | – | – | $7.39 \times 10^{-9}$ | 1.14 (1.08–1.20) | intron |
| | | | | | EA | 0.522 | 0.491 | $2.02 \times 10^{-5}$ | 1.10 (1.05–1.15) | |
| | | | | | AA | 0.083 | 0.075 | $9.97 \times 10^{-2}$ | 1.13 (0.98–1.30) | |
| | | | | | HA | 0.288 | 0.282 | $2.51 \times 10^{-5}$ | 1.25 (1.13–1.38) | |
| rs2041670 | 16p13 | 11174652 | CLEC16A[1] | T | Meta | – | – | $2.14 \times 10^{-16}$ | 0.85 (0.82–0.89) | intron |
| | | | | | EA | 0.287 | 0.316 | $2.34 \times 10^{-12}$ | 0.84 (0.80–0.88) | |
| | | | | | AA | 0.536 | 0.564 | $2.78 \times 10^{-3}$ | 0.89 (0.83–0.96) | |
| | | | | | HA | 0.201 | 0.251 | $1.67 \times 10^{-3}$ | 0.84 (0.75–0.94) | |
| rs223889 [d] | 16q13 | 57392241 | PLLP-CCL22 | T | Meta | – | – | $1.08 \times 10^{-8}$ | 1.21 (1.13–1.29) | near-gene-5 |
| | | | | | EA | 0.307 | 0.285 | $1.16 \times 10^{-4}$ | 1.13 (1.06–1.20) | |
| | | | | | AA | 0.697 | 0.685 | $5.56 \times 10^{-3}$ | 1.29 (1.08–1.55) | |
| | | | | | HA | 0.429 | 0.381 | $3.22 \times 10^{-4}$ | 1.28 (1.12–1.47) | |
| rs1749792 | 16q22 | 68569440 | ZFP90[1] | T | Meta | – | – | $3.66 \times 10^{-11}$ | 1.14 (1.10–1.19) | |
| | | | | | EA | 0.253 | 0.227 | $4.32 \times 10^{-6}$ | 1.13 (1.07–1.18) | |
| | | | | | AA | 0.320 | 0.274 | $1.64 \times 10^{-6}$ | 1.21 (1.12–1.30) | |
| | | | | | HA | 0.298 | 0.273 | $4.50 \times 10^{-2}$ | 1.11 (1.00–1.23) | |
| rs8072449 | 17q25 | 73312184 | GRB2 | G | Meta | – | – | $1.19 \times 10^{-11}$ | 0.84 (0.80–0.89) | |
| | | | | | EA | 0.159 | 0.164 | $1.08 \times 10^{-6}$ | 0.86 (0.81–0.91) | |
| | | | | | AA | 0.804 | 0.835 | $6.76 \times 10^{-6}$ | 0.79 (0.71–0.88) | |
| rs13344313 | | | | | HA | 0.168 | 0.193 | $2.97 \times 10^{-2}$ | 0.88 (0.78–0.99) | |
| | 19p13 | 18517767 | LRRC25-SSBP4 | A | Meta | – | – | $1.07 \times 10^{-8}$ | 0.90 (0.86–0.93) | |
| | | | | | EA | 0.259 | 0.29 | $7.03 \times 10^{-7}$ | 0.88 (0.84–0.93) | |
| | | | | | AA | 0.391 | 0.407 | $8.50 \times 10^{-2}$ | 0.93 (0.86–1.01) | |
| | | | | | HA | 0.232 | 0.257 | $1.45 \times 10^{-2}$ | 0.88 (0.79–0.97) | |
| rs56154925 | 19q13 | 55737798 | PTPRH-TMEM86B | T | Meta | – | – | $2.27 \times 10^{-8}$ | 0.88 (0.84–0.92) | near-gene-3 |
| | | | | | EA | 0.164 | 0.178 | $1.17 \times 10^{-5}$ | 0.88 (0.83–0.93) | |
| | | | | | AA | 0.187 | 0.207 | $9.51 \times 10^{-3}$ | 0.88 (0.80–0.97) | |
| | | | | | HA | 0.200 | 0.218 | $2.02 \times 10^{-2}$ | 0.88 (0.78–0.98) | |
| rs137956 | 22q13 | 40293463 | ENTHD1-GRAP2 | G | Meta | – | – | $5.00 \times 10^{-8}$ | 0.88 (0.84–0.92) | |
| | | | | | EA | 0.424 | 0.446 | $3.36 \times 10^{-4}$ | 0.92 (0.88–0.96) | |
| | | | | | AA | 0.099 | 0.114 | $8.50 \times 10^{-3}$ | 0.85 (0.75–0.96) | |
| | | | | | HA | 0.475 | 0.502 | $2.75 \times 10^{-4}$ | 0.84 (0.77–0.92) | |
| *Tier 2 Meta-Analysis* | | | | | | | | | | |
| rs6662618 [d] | 1p22 | 92935411 | GFI1-EVI5 | T | Meta | – | – | $1.02 \times 10^{-7}$ | 1.14 (1.08–1.21) | |
| | | | | | EA | 0.182 | 0.157 | $1.54 \times 10^{-6}$ | 1.18 (1.10–1.26) | |
| | | | | | AA | 0.450 | 0.442 | $1.18 \times 10^{-1}$ | 1.10 (0.98–1.23) | |
| | | | | | HA | 0.250 | 0.229 | $5.55 \times 10^{-2}$ | 1.14 (1.00–1.29) | |

**Table 3 (Continued ).**

| SNP | Chr. | Position (b37) | Gene region* | Ref. allele | Ancestry | RAF case | RAF control | P value | OR (95% CI) | dbSNP function[†] |
|---|---|---|---|---|---|---|---|---|---|---|
| rs835573 | 1p12 | 120464165 | NOTCH2 | A | Meta | – | – | $2.23 \times 10^{-7}$ | 1.13 (1.08–1.18) | intron |
| | | | | | EA | 0.138 | 0.122 | $2.74 \times 10^{-5}$ | 1.15 (1.08–1.22) | |
| | | | | | AA | 0.438 | 0.415 | $1.57 \times 10^{-2}$ | 1.10 (1.02–1.19) | |
| | | | | | HA | 0.180 | 0.159 | $6.44 \times 10^{-2}$ | 1.12 (0.99–1.27) | |
| rs12068671 | 1q24 | 172681031 | FASLG | G | Meta | – | – | $6.66 \times 10^{-8}$ | 0.88 (0.84–0.92) | |
| | | | | | EA | 0.171 | 0.186 | $5.92 \times 10^{-6}$ | 0.88 (0.83–0.93) | |
| | | | | | AA | 0.383 | 0.393 | $2.47 \times 10^{-1}$ | 0.96 (0.88–1.03) | |
| | | | | | HA | 0.104 | 0.138 | $1.39 \times 10^{-3}$ | 0.79 (0.69–0.91) | |
| rs7579944 | 2p23 | 30445026 | LBH[1] | A | Meta | – | – | $5.11 \times 10^{-8}$ | 0.90 (0.86–0.93) | |
| | | | | | EA | 0.340 | 0.364 | $2.03 \times 10^{-4}$ | 0.92 (0.88–0.96) | |
| | | | | | AA | 0.698 | 0.727 | $5.49 \times 10^{-4}$ | 0.86 (0.79–0.94) | |
| | | | | | HA | 0.340 | 0.367 | $1.76 \times 10^{-2}$ | 0.89 (0.81–0.98) | |
| rs461193 | 5p15 | 1368997 | BC034612 | G | Meta | – | – | $1.20 \times 10^{-7}$ | 1.13 (1.08–1.18) | intron |
| | | | | | EA | 0.214 | 0.187 | $1.18 \times 10^{-5}$ | 1.13 (1.07–1.19) | |
| | | | | | AA | 0.255 | 0.241 | $9.94 \times 10^{-2}$ | 1.08 (0.99–1.18) | |
| | | | | | HA | 0.169 | 0.162 | $7.95 \times 10^{-3}$ | 1.19 (1.05–1.35) | |
| rs909788 | 6p22 | 16636461 | ATXN1[1] | A | Meta | – | – | $1.14 \times 10^{-7}$ | 1.10 (1.06–1.15) | intron |
| | | | | | EA | 0.455 | 0.432 | $3.26 \times 10^{-5}$ | 1.10 (1.05–1.15) | |
| | | | | | AA | 0.676 | 0.662 | $1.31 \times 10^{-1}$ | 1.06 (0.98–1.15) | |
| | | | | | HA | 0.432 | 0.402 | $8.56 \times 10^{-4}$ | 1.17 (1.07–1.29) | |
| rs11154801 | 6q23 | 135739355 | AHI1 | T | Meta | – | – | $3.60 \times 10^{-7}$ | 1.11 (1.06–1.16) | intron |
| | | | | | EA | 0.384 | 0.363 | $1.12 \times 10^{-5}$ | 1.11 (1.06–1.16) | |
| | | | | | AA | 0.115 | 0.104 | $5.36 \times 10^{-2}$ | 1.13 (1.00–1.28) | |
| | | | | | HA | 0.307 | 0.298 | $7.98 \times 10^{-2}$ | 1.09 (0.99–1.21) | |
| rs7795074 | 7p15 | 26742154 | SKAP2 | A | Meta | – | – | $1.93 \times 10^{-7}$ | 0.89 (0.86–0.93) | intron |
| | | | | | EA | 0.327 | 0.339 | $1.49 \times 10^{-3}$ | 0.93 (0.89–0.97) | |
| | | | | | AA | 0.301 | 0.328 | $3.43 \times 10^{-3}$ | 0.88 (0.81–0.96) | |
| | | | | | HA | 0.281 | 0.324 | $4.26 \times 10^{-4}$ | 0.84 (0.76–0.92) | |
| rs6601327 | 8p23 | 9395532 | TNKS | C | Meta | – | – | $1.46 \times 10^{-7}$ | 1.10 (1.06–1.14) | |
| | | | | | EA | 0.398 | 0.382 | $5.36 \times 10^{-6}$ | 1.11 (1.06–1.16) | |
| | | | | | AA | 0.517 | 0.493 | $1.35 \times 10^{-2}$ | 1.10 (1.02–1.19) | |
| | | | | | HA | 0.526 | 0.492 | $2.21 \times 10^{-1}$ | 1.06 (0.97–1.16) | |
| rs17630235 | 12q24 | 112591686 | TRAFD1—C12orf51 | T | Meta | – | – | $2.50 \times 10^{-7}$ | 1.12 (1.06–1.18) | intron |
| | | | | | EA | 0.457 | 0.422 | $2.82 \times 10^{-5}$ | 1.10 (1.05–1.15) | |
| | | | | | AA | 0.069 | 0.063 | $1.91 \times 10^{-1}$ | 1.11 (0.95–1.30) | |
| | | | | | HA | 0.256 | 0.257 | $1.83 \times 10^{-3}$ | 1.18 (1.06–1.31) | |

Novel regions have not previously been identified by SNP associations with P values $<5 \times 10^{-8}$ and are highlighted in gray
Regions that are the first observed associations in these ancestries are indicated with a superscript [1] in the gene region.
*Named by the genes bounding the region of association, unless literature strongly implicated a specific gene;
[d] or [r]: Dominant, or recessive model; if not noted, additive model was used;
[†]dbSNP's predicted functional effect.

least a subset of the alleles is greater when the overall genetic load is high. HA and AA showed markedly smaller ORs (between 3 and 10), reflecting the reduced predictive ability of EA-identified SLE risk loci in non-EA populations and the lack of capturing non-EA SLE risk loci on the Immunochip.

The total non-HLA weighted genetic load was correlated with an earlier age at SLE diagnosis in EA ($r_{Spearman} = -0.14$, $P = 0.0001$), and HA ($r_{Spearman} = -0.10$, $P = 0.0012$), but not AA ($r_{Spearman} = 0.04$, $P = 0.54$). Kaplan–Meier curves in the EA showed separation accelerates at ~35 years (Supplementary Fig. 18). The HLA-based genetic load was not correlated with age of onset ($P > 0.05$) in any ancestry.

**Mapping SNP associations to eQTLs.** Many SLE-associated SNPs are, or are in LD with, *cis* eQTLs (Supplementary Data 12 and Supplementary Figs 13–16) and potentially link associations with specific genes. In ancestry-specific eQTL analyses (Supplementary Data 12), EA yielded 96 unique SNPs or their proxies mapping to 193 unique genes, followed by HA (22 unique SNPs; 34 genes) and AA (10 unique SNPs; 17 genes). eQTL analyses based on the meta-analysis SNPs yielded 107 unique genes, identified by 40 SNPs (or their proxies), mostly from

whole blood, monocytes or B-cell derived LCL (Supplementary Data 12). Novel and previously implicated SLE genes were identified (for example, *BANK1*, *IRF5*). Interestingly, a number of SNPs were associated with expression levels for multiple genes. For example, four SNPs were associated with expression levels of at least three genes, and one SNP, newly associated in this study (rs8072449; 17q25), were associated with expression levels of eight genes. Thus, some associated SNPs, either directly or via LD with proxy SNPs, contribute to disease by modifying expression levels of multiple genes, potentially through transcription binding sites. Supplementary Data 13 and 14 provide predicted functional characterization of the 206 SNPs from Tiers 1 to 2 that are in RegulomeDB and HaploReg. These predictions are informative for generating hypotheses that can be experimentally tested.

## Discussion

Applying the Immunochip to these multi-ancestral SLE case-control samples has identified 24 novel SLE-risk regions, replicated established SLE-risk loci and extended their impact into other ancestries, and refined association signals via

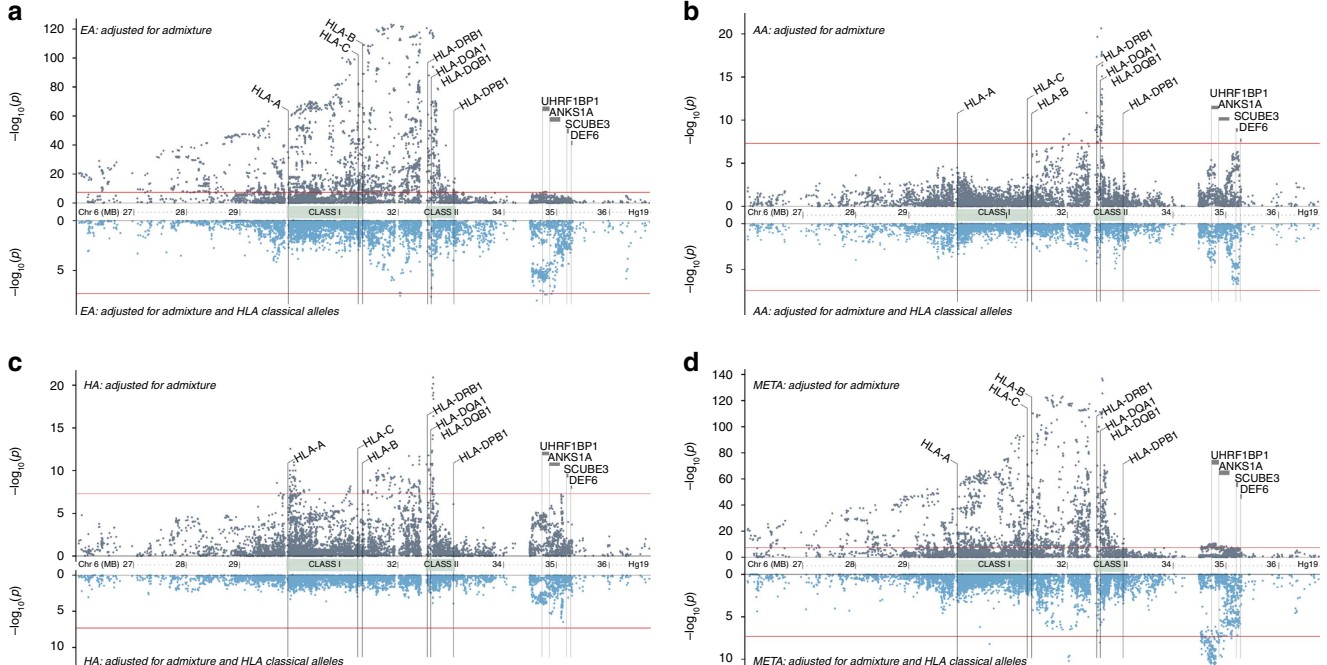

**Figure 2 | HLA SNP associations with and without adjustment of classical HLA alleles.** SNPs spanning the extended MHC region showed significant associations across (**a**) European ancestry, (**b**) African American, (**c**) Hispanic ancestry, and the (**d**) meta-analysis. The classical HLA alleles, from the ethnic-specific stepwise-models (Supplementary Data 5), accounted for a majority of the MHC SNP signals. For each plot, the Tier 1 threshold, $P \leq 5 \times 10^{-8}$, is indicated by the red line. Associations, downstream in 6p21 spanning UHRF1BP1-DEF6 were largely unaffected after adjusting for classical HLA alleles and appear independent of the MHC.

transancestral mapping. Over 50% of associated regions had multiple independent SNP associations. Many of these associations were linked via eQTL analysis to specific genes, a process that can accelerate discovery of critical pathways. The contrast of associations and genes across ancestries documents numerous ethnic-specific associations the ancestral diversity in SLE etiology; for example, HA regions not showing equivalent associations in EA include 3p11 (*EPHA3-PROS1*), 6q25 (*RSPH3*), 12q15 (*DYRK2-IFNG*), 12q21 (*SYT1*), 14q31 (*GALC*), 16q21 (*CSNK2A2-CCDC113*) and 22q12 (*C1QTNF6*). In total, these results underscore the shared and distinct genetic profiles of SLE relative to other autoimmune diseases.

To understand disease biology and prevalence across populations, distinguishing shared versus ancestry-specific associations is important because an allele identified in one population is likely relevant in others[33]. Clustering by allele frequencies in cases and comparing risk allele admixture estimates, three clusters emerged: (1) alleles with comparable frequencies across populations without strong deviations in average admixture, (2) alleles with increased AA-ancestral contribution and (3) alleles with reduced AA-ancestral contribution and increased CEU admixture. The increased European ancestry observed in less common AA risk alleles likely reflects complex demographic histories and admixture patterns.

The nonlinear nature of how genetic load affects SLE risk leads us to posit the *cumulative hit hypothesis for autoimmune diseases*. That is, in our current environment the immune system can absorb, with a modest increase in risk, individual risk polymorphisms. But as the number of risk variants increases, the system becomes overwhelmed and immune dysregulation occurs. Currently, it is unclear whether it is the entire genetic load or only a subset of variants driving the nonlinear association. In addition, increasing genetic load correlates with an earlier age of disease onset. These hypotheses are testable within specific

and across autoimmune diseases given their shared genetic architecture.

Despite the large sample size, there was no robust evidence for SNP-gender, SNP–SNP or SNP–HLA allele interactions, suggesting that pairwise-interactions among these Immunochip loci are not a major source of missing heritability. While the lack of pairwise interactions across the immune-centric loci may be surprising given the statistical power of the study, the current analysis does not preclude higher-order interactions; albeit agnostic scans for such interactions are analytically challenging. Furthermore, given the nonlinear effect of genetic load on risk, explicit and strong pairwise interactions may not be the correct hypothesis—gene-based or pathway-based interactions may be more important. Because of limitations in the data, gene-environment interactions were not computed and this area needs study.

The individual roles of DR3 and DR15 haplotypes in SLE risk are well-established. However, in all three ancestries, having two different risk alleles yielded higher SLE risk than having two copies of the same risk allele. This is similar to type 1 diabetes, where heterozygotes for type 1 diabetes-associated haplotypes, DR3 and DR4, have shown higher risk of disease. It is hypothesized that this effect is driven via formation of DQA1 and DQB1 trans-heterodimers. In contrast, SLE risk alleles in DR3 and DR15 stem from divergent ancient haplotypes[18]; likewise, trans-pairing has not been shown between DQA and DQB in these two haplotypes[34,35].

Due to the highly polymorphic nature of HLA alleles and their protein products, it is important to consider high-order relationships among amino acids in three-dimensional space[36]. Standard regression techniques using amino acids in isolation can be problematic and inappropriate for inference[37]. To account for higher-order relationships among amino acids, we (1) clustered alleles by protein sequence similarity, (2) compared associations within and between clusters and (3) identified, when possible,

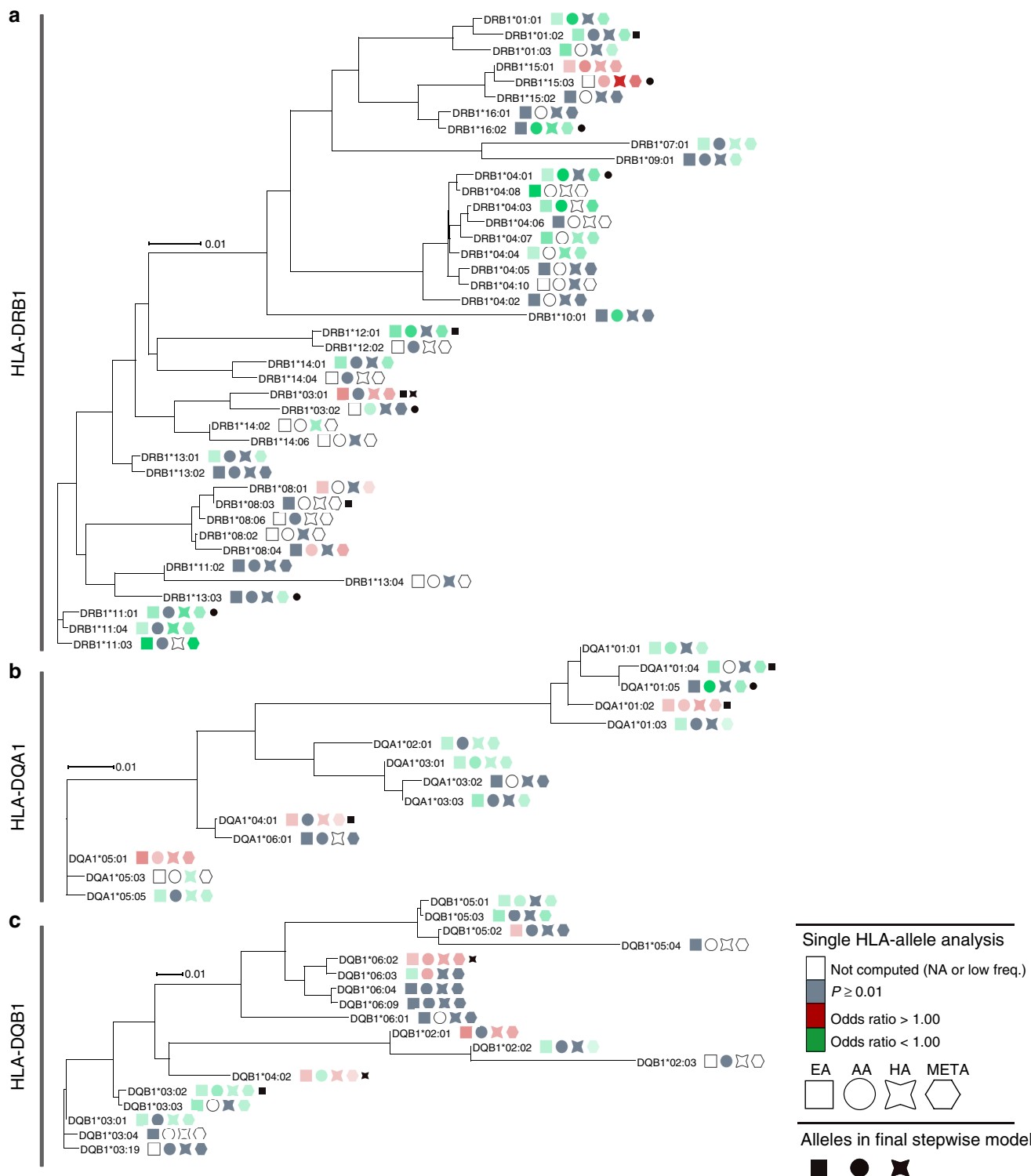

**Figure 3 | Clustering of HLA Class II alleles by amino acid sequence similarity.** For (**a**) DRB1, (**b**) DQA1, and (**c**) DQB1, the odds ratios for each cohort are superimposed on the cluster if the SLE association *P*-value was less than 0.01. Alleles that were present in the multi-locus model from the stepwise procedure are also denoted. This process aims to identify clusters with shared SLE risk or not-risk odds ratios across the three cohorts. Such clusters help identify potential amino acid sequences contributing to SLE risk. For example, DRB1*15:01 and 15:03 are clustered amongst protective alleles, suggesting presence of specific amino acids differentiating risk (Supplementary Figs 8 and 9).

amino acids that uniquely distinguished the risk alleles. This approach identified several examples of specific amino acids differentiating risk and protective HLA alleles. For example, the DRB15*01 amino acids −1, 47 and 71 were unique to risk alleles. The combination of Ala71 and Phe47 create a hydrophobic space in the protein binding pocket compared to the alternatives

observed (Glu71 and Tyr47; or Arg71 and Tyr47). In addition to antigen binding, there is a vast array of HLA allele-specific properties, including surface expression stability[35], influence of DNA methylation[38] and DR-DQ heterodimers[39]. Such findings may help prioritize functional experiments, as we work towards understanding the HLA mechanisms of SLE.

**Table 4 | Tier 1 non-HLA meta-analysis regions noted for transracial mapping.**

| Gene region* | SNP | Chr. | Position (b37) | Ref. allele | Ancestry | RAF case | RAF control | P value | OR (95% CI) | dbSNP function† |
|---|---|---|---|---|---|---|---|---|---|---|
| TNFSF4-LOC100506023 | rs2205960 | 1q25 | 173191475 | A | Meta | – | – | $1.16 \times 10^{-30}$ | 1.30 (1.23–1.38) | |
| | | | | | EA | 0.267 | 0.225 | $3.84 \times 10^{-23}$ | 1.29 (1.23–1.36) | |
| | | | | | AA | 0.066 | 0.050 | $7.46 \times 10^{-4}$ | 1.33 (1.13–1.56) | |
| | | | | | HA | 0.390 | 0.314 | $2.01 \times 10^{-7}$ | 1.29 (1.17–1.43) | |
| TNFSF4-LOC100506023 | rs1539255 | 1q25 | 173322660 | T | Meta | – | – | $1.60 \times 10^{-19}$ | 0.84 (0.81–0.87) | intron |
| | | | | | EA | 0.311 | 0.350 | $2.37 \times 10^{-12}$ | 0.85 (0.81–0.89) | |
| | | | | | AA | 0.428 | 0.460 | $7.24 \times 10^{-4}$ | 0.88 (0.81–0.95) | |
| | | | | | HA | 0.273 | 0.326 | $1.06 \times 10^{-6}$ | 0.78 (0.71–0.86) | |
| NMNAT2-SMG7-NCF2‡ | rs17484292 | 1q25 | 183300050 | T | Meta | – | – | $9.97 \times 10^{-38}$ | 1.59 (1.40–1.79) | intron |
| | | | | | EA | 0.089 | 0.048 | $1.48 \times 10^{-39}$ | 1.77 (1.63–1.93) | |
| | | | | | AA | 0.012 | 0.009 | $2.07 \times 10^{-1}$ | 1.27 (0.88–1.85) | |
| | | | | | HA | 0.051 | 0.035 | $3.07 \times 10^{-5}$ | 1.61 (1.29–2.02) | |
| NMNAT2-SMG7-NCF2 | rs10911363 | 1q25 | 183549757 | A | Meta | – | – | $2.52 \times 10^{-17}$ | 1.17 (1.13–1.22) | intron |
| | | | | | EA | 0.317 | 0.275 | $1.18 \times 10^{-13}$ | 1.19 (1.14–1.25) | |
| | | | | | AA | 0.329 | 0.319 | $2.40 \times 10^{-1}$ | 1.05 (0.97–1.14) | |
| | | | | | HA | 0.399 | 0.333 | $3.73 \times 10^{-7}$ | 1.27 (1.16–1.39) | |
| IL2-IL21 | rs11724582 | 4q27 | 123391464 | C | Meta | – | – | $1.71 \times 10^{-8}$ | 0.88 (0.84–0.93) | |
| | | | | | EA | 0.262 | 0.280 | $2.49 \times 10^{-6}$ | 0.89 (0.84–0.93) | |
| | | | | | AA | 0.129 | 0.152 | $9.86 \times 10^{-4}$ | 0.83 (0.75–0.93) | |
| | | | | | HA | 0.142 | 0.166 | $3.58 \times 10^{-1}$ | 0.94 (0.83–1.07) | |
| PTTG1-MIR146A | rs2431098 | 5q33 | 159887336 | C | Meta | – | – | $3.29 \times 10^{-21}$ | 1.19 (1.14–1.23) | |
| | | | | | EA | 0.532 | 0.497 | $5.09 \times 10^{-13}$ | 1.17 (1.12–1.23) | |
| | | | | | AA | 0.404 | 0.351 | $1.49 \times 10^{-8}$ | 1.25 (1.16–1.36) | |
| | | | | | HA | 0.470 | 0.441 | $4.62 \times 10^{-3}$ | 1.14 (1.04–1.25) | |
| IRF5-TNPO3 | rs4728142 | 7q32 | 128573967 | T | Meta | – | – | $3.38 \times 10^{-84}$ | 1.44 (1.39–1.50) | |
| | | | | | EA | 0.531 | 0.446 | $6.21 \times 10^{-51}$ | 1.40 (1.34–1.46) | |
| | | | | | AA | 0.327 | 0.264 | $1.16 \times 10^{-12}$ | 1.35 (1.24–1.47) | |
| | | | | | HA | 0.518 | 0.394 | $2.10 \times 10^{-27}$ | 1.65 (1.51–1.81) | |
| IRF5-TNPO3 | rs35000415 | 7q32 | 128585616 | T | Meta | – | – | $1.17 \times 10^{-99}$ | 1.82 (1.69–1.96) | intron |
| | | | | | EA | 0.184 | 0.118 | $5.67 \times 10^{-70}$ | 1.73 (1.63–1.84) | |
| | | | | | AA | 0.040 | 0.022 | $1.80 \times 10^{-7}$ | 1.86 (1.47–2.34) | |
| | | | | | HA | 0.298 | 0.158 | $7.11 \times 10^{-33}$ | 1.98 (1.77–2.22) | |
| LYN-RPS20 | rs2953898 | 8q12 | 56980803 | A | Meta | – | – | $4.43 \times 10^{-8}$ | 0.84 (0.79–0.90) | untranslated-3 |
| | | | | | EA | 0.218 | 0.226 | $6.89 \times 10^{-5}$ | 0.90 (0.85–0.95) | |
| | | | | | AA | 0.029 | 0.038 | $1.01 \times 10^{-2}$ | 0.77 (0.63–0.94) | |
| | | | | | HA | 0.124 | 0.167 | $3.60 \times 10^{-3}$ | 0.82 (0.72–0.94) | |
| PDHX-CD44 | rs353592 | 11p13 | 35119482 | T | Meta | – | – | $1.35 \times 10^{-8}$ | 0.89 (0.85–0.93) | |
| | | | | | EA | 0.441 | 0.475 | $4.89 \times 10^{-6}$ | 0.90 (0.86–0.94) | |
| | | | | | AA | 0.113 | 0.127 | $4.14 \times 10^{-2}$ | 0.88 (0.79–1.00) | |
| | | | | | HA | 0.286 | 0.339 | $4.89 \times 10^{-3}$ | 0.87 (0.79–0.96) | |
| SLC15A4 | rs1059312 | 12q24 | 129278864 | G | Meta | – | – | $6.53 \times 10^{-10}$ | 1.12 (1.07–1.16) | coding-synon |
| | | | | | EA | 0.423 | 0.392 | $6.26 \times 10^{-7}$ | 1.12 (1.07–1.17) | |
| | | | | | AA | 0.497 | 0.461 | $1.84 \times 10^{-4}$ | 1.16 (1.07–1.25) | |
| | | | | | HA | 0.615 | 0.571 | $2.17 \times 10^{-1}$ | 1.06 (0.97–1.16) | |
| NCOA5-CD40 | rs4810485 r | 20q13 | 44747947 | A | Meta | – | – | $9.95 \times 10^{-9}$ | 1.43 (1.17–1.76) | intron |
| | | | | | EA | 0.275 | 0.248 | $1.01 \times 10^{-7}$ | 1.38 (1.23–1.56) | |
| | | | | | AA | 0.079 | 0.080 | $2.16 \times 10^{-1}$ | 1.53 (0.78–3.02) | |
| | | | | | HA | 0.216 | 0.208 | $2.37 \times 10^{-2}$ | 1.42 (1.05–1.92) | |

The corresponding plots can been found in Supplementary Fig. 15.
d or r: Dominant, or recessive model; if not noted, additive model was used.
*Named by the genes bounding the region of association, unless literature strongly implicated a specific gene.
†dbSNP's predicted functional effect.
‡SNP's association is not supported by LD SNPs. Cluster call plot was verified for quality control. Additional verification of association will be required.

Two major limitations of this study are the comparably fewer non-EA SLE cases and appropriate controls, and the strong EA bias in the Immunochip content. Power calculations using allele frequencies and ORs from EA, and the number of AA cases and controls, yielded 445.5 expected Tier 1 and 2 SNP associations; however, only 64 were observed. Although differences in LD contribute to this result, the highly reduced number of detected associations relative to expected, plus the genetic load analyses, strongly suggest that ancestry-specific and -independent loci contribute to SLE risk. It is imperative to recruit more non-EA populations for genetic studies.

In conclusion, SLE has a strong genetic contribution to risk with ancestry-dependent and ancestry-independent contributions. SLE risk has shared and independent genetic contributions relative to other autoimmune diseases. This genetic risk manifests itself as a nonlinear function of the cumulative risk allele load, a pattern potentially shared across autoimmune and non-autoimmune diseases.

## Methods
**Study cohort.** Multiple studies provided de-identified DNA samples with approval from their respective institutional review boards or ethics committees. These ethics review committees included: Cedars-Sinai Medical Center

**Table 5 | Genetic Load and SLE risk.**

| | non-HLA SNPs from EA Top Hits* (N = 545) | | | HLA classical alleles from EA top hits* (N = 10) | | | Combined (N = 555) | | |
|---|---|---|---|---|---|---|---|---|---|
| | P value | OR (95% CI)† | c-statistic‡ | P value | OR (95% CI)† | c-statistic‡ | P value | OR (95% CI)† | c-statistic‡ |
| Un-Weighted | | | | | | | | | |
| AA | $5.31 \times 10^{-30}$ | 1.15 (1.13–1.18) | 0.591 (0.590) | $5.75 \times 10^{-7}$ | 2.44 (1.72–3.45) | 0.540 (0.538) | $2.84 \times 10^{-31}$ | 1.13 (1.11–1.16) | 0.593 (0.592) |
| EA§ | $6.15 \times 10^{-62}$ | 1.23 (1.20–1.26) | 0.694 (0.657) | $7.20 \times 10^{-41}$ | 5.15 (4.05–6.55) | 0.670 (0.608) | $5.24 \times 10^{-69}$ | 1.24 (1.21–1.28) | 0.702 (0.665) |
| HA | $1.76 \times 10^{-26}$ | 1.14 (1.12–1.17) | 0.647 (0.624) | $9.39 \times 10^{-20}$ | 3.41 (2.62–4.44) | 0.638 (0.560) | $7.02 \times 10^{-30}$ | 1.15 (1.12–1.18) | 0.651 (0.630) |
| *Weighted by Natural Log of the Odds Ratio (OR)* | | | | | | | | | |
| AA | $2.82 \times 10^{-36}$ | 2.59 (2.23–3.00) | 0.602 (0.601) | $9.34 \times 10^{-11}$ | 24.11 (9.20–63.17) | 0.551 (0.550) | $1.80 \times 10^{-40}$ | 2.71 (2.34–3.14) | 0.608 (0.607) |
| EA§ | $1.37 \times 10^{-101}$ | 7.83 (6.49–9.46) | 0.738 (0.714) | $7.60 \times 10^{-45}$ | 29.06 (18.16–46.52) | 0.678 (0.618) | $2.19 \times 10^{-121}$ | 8.48 (7.09–10.15) | 0.759 (0.734) |
| HA | $2.09 \times 10^{-46}$ | 3.76 (3.14–4.51) | 0.674 (0.660) | $8.61 \times 10^{-24}$ | 34.33 (17.24–68.37) | 0.645 (0.582) | $1.98 \times 10^{-57}$ | 4.26 (3.57–5.09) | 0.687 (0.675) |

*Top hits from EA sample without validation set of 2,000 SLE cases, 2,000 controls.
†Per 5 alleles.
‡Whole model statistic, and in parentheses, the c-statistic for model without admixture factors.
§EA random sample with 2,000 SLE cases, 2,000 controls.

Institutional Review Board; Central Ethic Committee of Denmark; Centrala etikprövningsnämnden; Comité de Etica de la Investigación de Centro Hospital Universitario Virgen Macarena; Centro de Estudios Reumatológicos. Santiago de Chile; Centro Hospitalar Universitário do Porto, Unidade de Imunologia Clinica e Comissão de Ética; CEPI (Comite de Etica de Protocolos de Investigacion) Institution: Hospital Italiano de Buenos Aires; Cincinnati Children's Hospital Medical Center Institutional Review Board; Clinical Research Unit, Padua University-Hospital, and Ethics Committee, Province of Padua; Comitato Etico Interaziendale AOU Maggiore della Carità Ethics Committee, Novara, Italy; Comite de Bioetica del Consejo Superior de Investigaciones Científicas; Comité de Docencia e Investigación, Hospital Escuela Eva Perón, Gro Baigorria, Santa Fe, Argentina; Comité de Docencia e Investigación, Sanatorio Parque SA; Comite de etica de la investigacion del HIGA San Martín de La Plata, Argentina; Comité de Ética en Investigación Instituto Nacional de Ciencias Médicas y Nutrición Salvador Zubirán; Comité de Ética en Investigación, Instituto Nacional de Medicina Genómica, Mexico; Comité de Ética en Investigación; Comité de Investigación de la Facultad de Medicina de la UANL y Hospital Universitario 'Dr José Eleuterio González'; Comite Docencia e Investigacion H.I.G.A. Dr Oscar Alende Mar del Plata; Comitè Ètic d'Investigació Clínica de l'Hospital Clínic de Barcelona; Comités de Ética, Bio Ética y de Investigación. Hospital G. Almenara, Esalud, Lima, Perú; Comites de Ética, Bioetica y de Investigación Hospital Nacional Guillermo Almenara Irigoyen, Lima-Perú;
Commission d'Ethique Hospitalo-Facultaire de l'Université catholique de Louvain; Duke University Health System Institutional Review Board; Ethics and Research Committee of Hospital General De Occidente; Fundacion Docencia e Investigacion Hopsital Italiano de Cordoba; Institution of Public Health and Clinical Medicine, Rheumatology, Umeå University, Umeå, Sweden; Institutional Review Board of the University of Puerto Rico Medical Sciences Campus; Institutional Review Board Office Northwestern University; Johns Hopkins University School of Medicine Institutional Review Board; London Central Research Ethics Committee Study sponsor: King's College London; Medical Ethical Committee (METc) of the University Medical Center Groningen; Medical University of South Carolina Institutional Review Board for Human Research; Northwell Health Human Research Protection Program; Oklahoma Medical Research Foundation Institutional Review Board; omisión Nacional de Investigación Científica y Comisión de Ética en Investigación en Salud, Instituto Mexicano del Seguro Social, México; Regional Ethical Review Board at Karolinska Institutet, Stockholm, Sweden; Regional Ethics Review Board in Linköping; Regional Human Medical Research Ethics Committee of the University of Szeged; SickKids REB; The Institution Review Boards for human research at UCLA; The Local Ethics Committee of the Karolinska University Hospital/Karolinska Institutet, Stockholm Sweden; The University Health Network, Research Ethics Board; Institutional Review Board for Human Use University of Alabama at Birmingham; UC Davis Institutional Review Board; UCSF Human Research Protection Program Institutional Review Board; UHN REB; University Health Network Research Ethics Board and by the local ethics boards of the CaNIOS investigators at the following centres: Montreal General Hospital, St Josephs' Heath Centre, Winnipeg Health Science Center, Queen Elizabeth II Health Sciences Centre, Ottawa Hospital, Hopital Notre-Dame, Calgary Health Sciences Centre, Centre Hospitalier Universitaire de Sherbrooke, and Hopital Maisonneuve-Rosemount; University Hospital of Gran Canaria Doctor Negrin Research Ethic Committee; University of Chicago Institutional Review Board; University of Southern California Health Sciences Institutional Review Board; University of Texas Southwestern Medical Center Institutional Review Board; Uppsala Ethical Review Board; Wake Forest University School of Medicine Institutional Review Board. All study participants provided written consent prior to study enrolment at the institution where the samples were collected. All SLE cases in this study were required to meet at least four of the eleven American College of Rheumatology classification criteria for SLE[40,41].

**Genotyping and quality controls.** Samples were genotyped on the custom-designed Immunochip Illumina Infinium Assay[9] according to Illumina's protocols, using the Illumina iScan scanner at the following centres: Oklahoma Medical Research Foundation, University of Texas Southwestern, HudsonAlpha Institute for Biotechnology, North Shore-LIJ Health System's Feinstein Institute for Medical Research. Intensity data were generated for all samples and sent to the Oklahoma Medical Research Foundation for genotype calling using OptiCall[42]. OptiCall default options were used with one exception: the '-nointcutoff' option was included to allow removal of intensity outliers. Subsequent genotype clusters were viewed against their intensity data using Evoker[43]. Genotype calling was completed in four batches, keeping samples genotyped at the same center in the same batch. Batches were designed to include samples of multiple ancestries when possible to improve rare variant calling. The ancestry breakdown for the batches was: Batch I was 15% European ancestry (EA), 7% African American ancestry (AA), 55% Asian ancestry (ASA), 23% Hispanic ancestry (HA); Batch II was 44% EA, 18% AA, 1.4% ASA, 36% HA; Batch III was 48% EA, 38% AA, 1% ASA, 13% HA; and Batch IV was 92% EA, 8% AA. Some samples called with the SLE Immunochip study samples were used for other Immunochip studies.

Samples were excluded if their call rates were <98% across SNPs that passed quality control filters. Duplicates and first-degree relatives were removed, retaining the sample with the highest call rate. The Immunochip does not have sufficient markers in the non-pseudoautosomal regions of chromosome X to reliably complete gender checks. Admixture estimates were computed using the program ADMIXTURE[44]. HapMap phase 2 individuals (CEU: Utah residents with ancestry from northern and western Europe; YRI: Yoruba in Ibadan, Nigeria; CHB: Han Chinese in Beijing, China) as anchoring populations. To facilitate testing for association between rare variants and SLE, and to improve multilocus modelling in regions of linkage disequilibrium (LD) among SNPs, a factor analysis was computed on the admixture estimates using principal component extraction and varimax rotation[45]. The resulting factors are orthogonal (independent) and thereby remove collinearity among the admixture estimates when used as covariates in linear models. Reduced collinearity should facilitate more robust analysis of rare variants. In addition, principal component (PC) analysis was computed using Eigensoft v4.2 (refs 46,47) including HapMap phase 2 individuals (CEU, YRI and CHB) as reference populations. Both the admixture and PC analyses were completed using a subset of SNPs generated by removing SNPs in LD ($r^2 > 0.2$), with minor allele frequency (MAF) <0.01, or with low call rate (<95%).

The admixture estimates and PCs were used to identify and remove genetic outliers. A SNP was removed from the primary analysis if it had an overall call rate <95%, exhibited significant differential missingness between cases and controls ($P < 0.05$), had significant departure from Hardy-Weinberg equilibrium expectations ($P < 1 \times 10^{-6}$ in cases, $P < 0.01$ in controls) or a cluster separation score <0.40. SNPs violating the above Hardy-Weinberg equilibrium thresholds were retained if there was convincing evidence of association at SNPs in linkage disequilibrium (LD) and the cluster plots indicated that the pattern was not due to poor genotype calling. Primary inference was based on SNPs with MAF ≥0.01. Finally, >10,000 SNP cluster plots were visually examined, including all SNPs reported, to remove results potentially based on poor genotyping.

To provide an estimate of the number of independent tests for multiple comparisons adjustment, the SNPs were LD pruned, $r^2 < 0.20$, within each ancestry. The union of these SNPs across ancestries was 46,744 uncorrelated SNPs, yielding a Bonferroni threshold of $P < 1.06 \times 10^{-6}$.

**Statistical analysis.** Regions in figures and tables are named by the genes bounding the regions of association or regions of significance for other statistical test, unless the literature strongly implicated a specific gene.

To test for an association between a SNP and case/control status within an ancestry, a logistic regression analysis was computed adjusting for admixture factors as covariates. Primary inference was based on the additive genetic model unless there was significant evidence of a lack-of-fit to the additive model ($P < 0.05$). If there was evidence of a departure from an additive model, then inference was based on the most significant of the dominant, additive, and recessive genetic models. The additive and recessive models were computed only if there were at least 10 and 30 individuals homozygous for the minor allele, respectively.

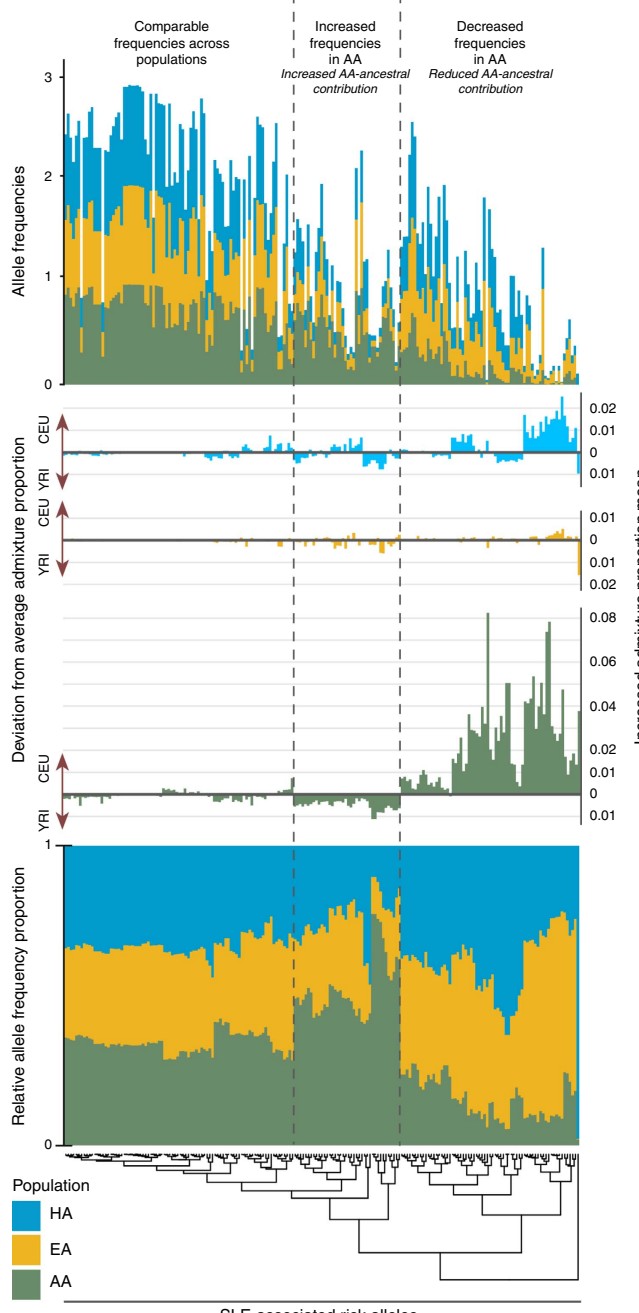

**Figure 4 | Ancestral landscape of SLE risk alleles.** Clustering by relative allele frequency yields three distinctive categories for SLE risk alleles: comparable frequencies across populations, increased frequencies in AA, and decreased frequencies in AA. The comparable frequency grouping contained the most risk alleles, of which, many were common alleles. This cluster had the smallest deviations from average admixture proportions, across the three cohorts. The increased frequencies in AA alleles exhibited moderate deviations towards greater AA-ancestral contribution. The largest deviations from average admixture were found within alleles exhibiting decreased frequencies in AA. These alleles were enriched for admixture deviations of increased CEU-ancestry. The patterns across relative allele frequencies reveal that ancestry-specific associations are largely not driven by monomorphic SNPs in other populations.

These tests of association were computed using the SNPGWA version 4.0 module of SNPLASH (https://www.phs.wakehealth.edu/public/bios/gene/downloads.cfm). For ancestry-specific analysis of the X chromosome, the data were first stratified by gender and then meta-analysed using the weighted inverse normal method

(weighted by sample size). The genomic control inflation factor ($\lambda_{GC}$) was calculated using a set of SNPs included on the Immunochip for a study investigating the genetic basis for reading and writing ability. The resulting $\lambda_{GC}$ was scaled to 1,000 cases and 1,000 controls to standardize comparisons across populations and studies.

Three tiers of statistical significance are reported. Tier 1 includes those SNPs that meet the literature-motivated genome-wide threshold of $5 \times 10^{-8}$. Tier 2 includes those SNPs that are not Tier 1 SNPs, but have a $P$ value for association less than $1 \times 10^{-6}$. Tier 3 includes those SNPs that do not meet criteria for Tiers 1 or 2, but meet a genome-wide Benjamini–Hochberg false discovery rate[48] adjusted $P$ value threshold of 0.05. The Tier 2 threshold meets the strict Bonferroni criteria for the number of uncorrelated SNPs ($r^2 < 0.20$).

Ancestry-specific logistic regression models were computed to test for evidence of interactions among all pairs of SNPs that had BH-FDR adjusted $P$ value $< 0.05$. Each logistic model contained the admixture factors, the two SNPs, and their centred cross-product term, with the latter term tested using the likelihood ratio test implemented in the Intertwolog module in SNPLASH. To adjust for the number of interactions tested, Bonferroni and BH-FDR adjusted $P$ values were computed. To test for ancestry-specific gender-by-SNP interactions, a case-only autosomal scan was computed; here, gender was the outcome and admixture factors and SNP were the predictors. To adjust for the number of tests computed, the BH-FDR adjusted $P$ values from the likelihood ratio test were computed for each SNP that passed quality control.

To determine how many distinct associations were within a genomic region, a manual stepwise procedure (that is, forward selection with backward elimination, entry and exit criteria of $P < 0.001$) was computed.

For the *transancestral meta-analyses*, three ancestries were examined for association and meta-analysed to better isolate shared SLE loci by leveraging their LD pattern differences. For each SNP, a nonparametric meta-analysis, weighted inverse normal method (weighted by sample size), was computed as implemented in METAL[49]. Regions of association were visually examined and tests of heterogeneity of the odds ratio were computed. Thus, for each region, ancestry-specific and meta-analytic tests of association and tests of heterogeneity are reported. The transancestral patterns of association and LD were visualized using LocusZoom[50]. Results from the weighted inverse normal method were compared to random effects meta-analyses and results of the regions were comparable.

Classical HLA alleles at HLA-A,-B,-C,-DPB1,-DQA1,-DQB1 and -DRB1 were imputed using the program HIBAG[51]. HIBAG uses an ensemble classifier and bagging technique to arrive at an average posterior probability. Unlike alternative imputation software such as BEAGLE[52], HLA*IMP[53] and SNP2HLA[54], HIBAG did not require training data for any of our three cohorts, as it provides multiple ancestry reference panels (European, African, Hispanic and Asian). This, combined with its accuracy rates being comparable to other approaches[51], made HIBAG an ideal method for HLA imputation in our EA, AA, and HA cohorts. To account for imputation uncertainty, the allele dosage was utilized for all analyses. To filter out the lowest frequency alleles, a minimum best guess allele count of 10 was required in either the cases or controls for each allele, in each cohort.

For analysis of classical HLA alleles, single-allele associations were evaluated using logistic regression under the additive model and accounting for imputation uncertainty via allelic dose. To account for population substructure, cohort-specific factors were used as covariates (EA: factors 1–4; AA: factors 1–3; HA: factors 1–2) in each analysis. Meta-analysis was completed for any allele that had a single-allele analysis in at least two cohorts. Evidence of association from each cohort was combined using the weighted inverse normal method via METAL[49] and tests for heterogeneity of the odds ratio were computed.

To build multi-locus ancestry-specific models of classical HLA alleles for case/control status of SLE, stepwise regression models were computed. Stepwise logistic modelling (forward selection with backward elimination) was computed using all of the classical HLA alleles that met the QC criteria, including requiring at least a count of 10 alleles from the best guess allele count cross the individuals within an ancestry. The entry and exit criteria were set to $P < 0.01$ for each of the three cohorts. As in the single-allele analysis, the logistic models tested for an additive effect of the alleles and accounted for imputation uncertainty via allelic dose.

To evaluate and compare classical HLA allele associations across the three cohorts, the results from the single-allele and multilocus modelling were visualized in the context of classical HLA protein sequence similarity. Protein sequences for all observed HLA-imputed alleles were retrieved from the EMBL-EBI Immunogenetics HLA Database[55]. Sequences within an HLA-gene were aligned using ClustalOmega[56]. Unrooted phylogenetic trees for each of the HLA loci were then generated by Clustal-W2 via the aligned amino acid sequences. The neighbour-joining method, a distance matrix method, utilized a Markov chain of nucleotide or amino acid substitution[57]. The neighbour-joining method uses this distance information to iteratively evaluate all pairings of neighbours in order to construct a tree that minimizes the branch length at each stage of clustering[58]. The resulting trees were visualized using Dendroscope[59]. All results from the single-allele and multilocus classical HLA associations from the three cohorts were graphically displayed on the unrooted trees.

A second set of ancestry-specific single-SNP analyses was computed across the HLA locus and surrounding region, while adjusting for the primary SLE-associated HLA risk alleles from the stepwise modelling. The logistic regression model was computed, as above, considering the fit to the three genetic models (dominant,

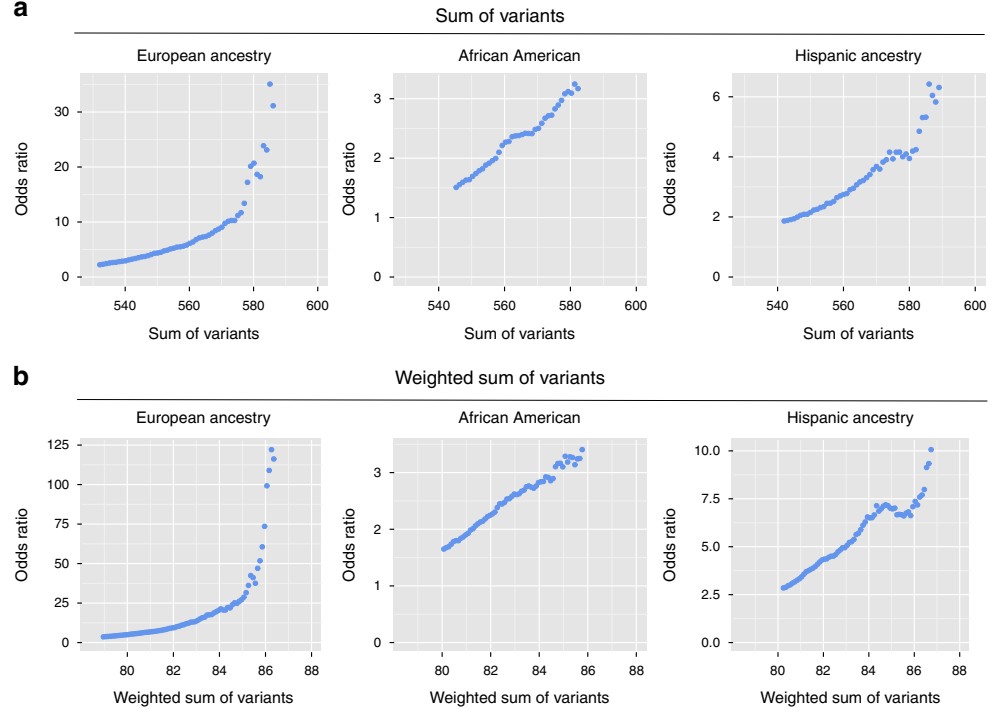

**Figure 5 | The non-additive effect of EA risk-allele genetic load on SLE risk.** The cumulative effect of EA SLE-risk alleles (cumulative hits) on an individual's risk of SLE is greater than if the individual SNPs were acting independently/additively. (**a**) The genetic load was computed as the sum of the number of EA risk variants from the Tier 1, 2 or 3 SNPs that met the region-specific stepwise modelling (see Online Methods). In the AA, HA and an independent set of 2,000 EA cases and 2,000 EA controls, the samples with the lowest 10% in risk-allele counts were identified and formed the baseline comparison group. Using a moving window of 10 in the allele count, the odds ratio for that window relative to the lowest 10% was computed and graphed. (**b**) The process was repeated for a weighted sum of the number of EA risk-allele variants. Here, the alleles are weighted by the natural logarithm of the odds ratio for that SNP's association with SLE. The corresponding moving window for the weighted genetic load used a window size of 3. Supplementary Fig. 17 plots the natural logarithm of the odds ratio (instead of the odds ratio) of genetic load versus SLE risk.

additive, recessive); the additive model required at least 10 homozygotes for the minor allele, while the recessive model required at least 30. The meta-analysis of these results was computed using METAL.

The Wald tests for HLA-by-SNP and HLA-by-gender interactions were computed using logistic regression models that adjusted for admixture factors and included both the main effects of the HLA and SNP (or gender) and their centred cross product as the multiplicative interaction term.

To test whether there was a difference in SLE risk between individuals homozygous for the same risk allele versus heterozygous for two different risk alleles, a Wald test from a logistic regression model was computed adjusting for admixture.

To examine ancestry of associated SLE risk alleles, genotyped SNPs from the population-specific (Tier1 and Tier 2) and the meta-analysis (primary and secondary) tables were compiled into a list of 205 unique SNPs. For evaluation, only SNPs of good quality across the three cohorts were retained. These criteria left 181 SNPs for comparison. In cases, admixture proportions of CEU and YRI were calculated using ADMIXTURE and then the average proportions were tallied for each cohort. Within each of the three populations and for each SNP, the risk allele's average admixture was computed. The resulting risk allele average admixture proportion was compared to the overall average sample admixture proportion in cases by computing the difference between risk allele and sample admixture proportion averages.

To evaluate the *SLE-risk allele genetic load*, the EA samples were partitioned into two groups: training (the entire EA sample minus 2,000 cases and 2,000 controls randomly chosen from the full EA cohort) and testing (the aforementioned 2,000 cases and 2,000 controls). In the training samples, the single SNP association and stepwise analyses were repeated to obtain a training set of SNPs that had BH-FDR adjusted *P*-value <0.05. From these results, the EA SLE-risk genetic load was calculated for each individual as the count of risk alleles from the training SNPs. Specifically, we define the EA SLE-risk allele genetic load as:

$$GRS_i = \sum_{k=1}^{N} \gamma_k RA_k,$$

where, $GRS_i$ is the genetic risk score for individual $i$; $\gamma_k$ is the beta coefficient for the *kth* SNP association with SLE and serves as the weight for that risk allele; $RA_k$ is the number of risk alleles for the *kth* SNP (0, 1, 2); and N is the number of SNPs. By definition of parameterizing relative to the risk allele, $\gamma_k > 0$ for all k. The EA SLE-risk genetic load was computed for AA, HA, and the EA testing samples. Individuals

whose genetic load (risk allele count) was in the lower 10% of the count distribution were used as the reference sample. A logistic regression model, including admixture factors as covariates, computed the odds ratio comparing the reference sample to samples within a moving window of 20 unweighted risk allele counts for the unweighted analysis and moving window of 4 for the weighted analysis). For example, a logistic model compared the risk of SLE for those in the lowest 10% to those whose risk allele counts ranged from 940 to 960 in the unweighted analysis. The next model and odds ratios were then computed, sliding the allele count up one (for example, 941–961). A plot of these odds ratios for moving windows of 20 counts was constructed to illustrate the pattern. The corresponding plot of the log(OR) = β from the genetic load association with SLE was generated to show that the nonlinearity was not due to the scale; that is, it documents a departure from linearity on the logit scale. A similar approach was completed for a weighted risk allele count, where each risk allele was weighted by the natural logarithm of the odds ratio from the EA SNP association analysis. Plots of the odds ratio effect of the EA genetic load (weighted and unweighted) were generated for AA, HA and the independent EA set.

Finally, for each ancestry an admixture-adjusted regression model was computed to test whether genetic load was associated with age of SLE onset. For ease of interpretation, the strength of the association was reported as the Spearman's rank correlation coefficient, but the *P* value is from the admixture-adjusted linear regression model.

**Functional annotation analysis.** To identify eQTLs for SLE-associated SNPs, all 1,000 Genomes SNPs in LD with the SLE-associated SNP were identified using SNAP[60]. Specifically, LD was computed using the CEU (for EA and HA) or YRI (for AA) data with an $r^2 \geqslant 0.5$ for Tier 1 and 2 SNPs. SNPs and their proxies were then queried in a data set downloaded from the eQTL Browser (http://eqtl.uchicago.edu/cgi-bin/gbrowse/eqtl/; Pritchard lab, University of Chicago) and the GTEx Portal (http://www.gtexportal.org). The eQTL Browser contains eQTL data surveyed from 17 eQTL studies, and the Blood eQTL Browser[61]. The GTEx Portal is a comprehensive resource, with eQTL data from 44 different tissues. When multiple proxies existed for the same eQTL (that is, same SNP and same gene), only the proxy with the lowest *P* value was retained.

RegulomeDB is a database that annotates SNPs with known and predicted regulatory elements (eQTLs, DNAase hypersensitivity, binding sites of transcription factors) in the intergenic regions of the human genome[22]. It includes high-throughput, experimental data sets from GEO, the ENCODE project,

published literature, as well as computational predictions and manual annotations to identify putative regulatory potential and identify functional variants[22]. The variants associated with SLE (identified in Tier 1 and 2 in any ancestry cohort) were queried in RegulomeDB.

HaploReg v2 is a tool for exploring annotations of the noncoding genome at variants on haplotype blocks[23] and uses LD information from the 1,000 Genomes Project Phase 1 individuals. It analyzes sets of SNPs for an enrichment of cell type-specific enhancers, and includes all dbSNP build 137 SNPs, predicted chromatin state in nine cell types, conservation across mammals, motif instances from ENCODE experiments, enhancer annotations on 90 cell types from the Roadmap Epigenome Mapping Consortium and eQTLs from the GTEx eQTL browser[23]. The query was performed using default settings, including LD calculations based on the 1,000 Genomes Phase 1 EUR individuals, and epigenome data from both the ENCODE and Roadmap Epigenome Mapping Consortium projects.

SNPs associated with SLE (Tiers 1 and 2) were annotated with the eQTL data and HaploReg v2 (ref. 23) to prioritize those with the highest biological potential. The top summary gene scores were summed across individual criteria (presence of an eQTL, presence of a nonsense or missense variant, promoter and enhancer status in a lymphoblastoid B-cell line (B-LCL), the presence of a DNase hypersensitivity site in any of five immune-related cell lines, presence of a conserved region, the presence of any bound protein, and transcription start site and enhancer status in any of 15 immune cell types), in the haplotype block of each SNP. In the calculation of the biological scores, each functional annotation was given a weight according to their regulatory potential. A score of '3' was given to SNPs in an LD block with any variant that mapped within an active or poised TSS in any of 15 immune cell types, was an eQTL, was non/missense, or mapped within an active promoter in a B-LCL. A score of '2' was given to SNPs in an LD block with any variant that mapped within an active upstream flanking TSS in any of 15 immune cell types or mapped within a conserved region. A score of '1' was given to SNPs in an LD block with any variant that mapped within a weak TSS or any enhancer in any of 15 immune cell types, mapped within a weak promoter or weak enhancer in a B-LCL, mapped within a DNase hypersensitivity site in any of 5 cell lines, or had any bound protein. The sum of these annotations resulted in a final biological score, ranging from zero to fifteen.

For each of the 146,111 (145,278 unique) SNPs that met quality control standards in at least one population, the flanking base pairs were identified using the UCSC reference genome (build 37). Once strand alignment was confirmed between the Immunochip and UCSC reference genome, it was evaluated whether either (or both) of a SNP's alleles created a CpG site in the 5′-3′ direction.

**Data availability.** The summary data are available at www.immunobase.org. Individual genotype data, consistent with the respective Institutional Review Board approval and subject consent, are available from the corresponding authors.

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

## Acknowledgements

We gratefully acknowledge the Alliance for Lupus Research for funding and support. The research was supported in part by awards from the Arthritis Research UK Special Strategic Award (ref. 19289) and from George Koukis (T.J.V.). In addition, the research was funded/supported by the National Institute for Health Research (NIHR) Biomedical Research Centre based at Guy's and St Thomas' NHS Foundation Trust and King's College London (T.J.V.). The work would not be possible without funding from the NIH grants AR049084 (RPK, EEB); the International Consortium on the Genetics of Systemic Lupus Erythematosus (SLEGEN) AI083194 (J.B.H.); CA141700, AR058621 Proyecto de Excelencia, Consejería de Andalucía (M.E.A.R.); AR043814 and AR-065626 (B.P.T.); AR060366, MD007909, AI107176 (S.K.N.); AR-057172 (C.O.J.); RC2 AR058959, U19 A1082714, R01 AR063124, P30 GM110766, R01 AR056360 (P.M.G.); P60 AR053308 (L.A.C.), MUSC part is from UL1RR029882 (G.S.G., D.L.K.) and 5P60AR062755 (G.S.G., D.L.K., P.R.R.). Oklahoma Samples U19AI082714, U01AI101934, P30GM103510, U54GM104938 and P30AR053483 (J.A.J., J.M.G.); Northwestern P60 AR066464 and 1U54TR001018 (R.R.G.); This study was supported by the US National Institute of Arthritis and Musculoskeletal and Skin Diseases of the National Institutes of Health (NIH) under Award Numbers K01 AR067280 and P60 AR062755 (PSR); N01AR22265 (funded collection of APPLE samples) (LES) and the APPLE Investigators; R01AR43727,NIH AR 043727 and 069572 (M.P.); NIAMS/NIH P50-AR055503 (D.R.K.).

We would like to also thank the RILITE foundation for financial support (C.D.L.). Additional funding for Immunochip genotyping was provided by Genentech.

## Author contributions

H.C.A., D.S.C.G. and J.A.K. contributed equally. P.M.G., R.R.G., C.D.L. and T.J.V. jointly supervised research. P.M.G., R.R.G., C.D.L., T.J.V., D.S.C.G., J.A.K., M.E.A., T.W.B., L.A.C., J.B.H., T.D.H., C.O.J., R.P.K., P.S.R., E.D.S., K.L.S., B.P.T. and E.K.W. conceived and designed the experiments. P.M.G., R.R.G., J.A.K., C.D.L., E.D.S., T.J.V. and E.K.W. performed experiments. H.C.A., M.E.C., T.D.H., J.A.K., C.D.L., M.C.M., D.R.M. and E.K.W. performed statistical analysis. H.C.A., M.E.C., D.S.C.G., T.D.H., K.M.K., J.A.K., L.C.K., C.D.L., M.C.M., D.R.M., P.S.R. analysed the data. P.M.G., R.R.G., R.P.K., C.D.L., E.D.S., T.J.V. and E.K.W. contributed reagents, materials, and analysis tools. H.C.A., M.E.C., P.M.G., R.R.G., T.D.H., C.D.L., M.C.M. and T.J.V. wrote the manuscript. E.M.A.-V., G.S.A., M.E.A., A.M.B., V.B., T.W.B., A.A.B., G.A.B., T.B., M.B., E.E.B., H.I.B., M.H.C., J.C.A., L.C., R.C., L.A.C., J.M.C.-V., S.D., B.M.D.S., S.R.D., I.D., A.D., J.C.E., E.E., J.A.E.-V., P.R.F., B.I.F., J.F., M.A.G., I.G., G.G., D.D.G., P.K.G., I.G.d.l.T., J.M.G., J.L.H., C.O.J., J.A.J., C.G.M.K., D.L.K., D.R.K., R.P.K., L.K., H.L., B.R.L., Q.Z.L., M.A.M., J.M., J.M.M., J.T.M., P.M., J.F.M., S.K.N., T.B.N., J.R.O., L.O., N.O., M.P., C.A.P., B.A.P., J.P., P.R., R.R., J.D.R., L.R., J.M.S., C.A.S., J.K.S., L.E.S., H.R.S., R.S., M.F.S., E.D.S., K.L.S., C.S., E.S., A.C.S., S.D.T., S.M.A.T., L.T., B.P.T., T.T., C.V., L.M.V., D.J.W., M.H.W. and J.E.W. contributed samples. E.M.A.-V., G.S.A., M.E.A., A.M.B., V.B., T.W.B., A.A.B., G.A.B., T.B., M.B., E.E.B., H.I.B., M.H.C., J.C.A., L.C., R.C., L.A.C., J.A.C., J.M.C.-V., D.S.C.G., S.D., B.M.D.S., S.R.D., I.D., A.D., J.C.E., E.E., J.A.E.-V., P.R.F., B.I.F., J.F., P.M.G., M.A.G., I.G.d.l.T, G.G., D.D.G., R.R.G., P.K.G., I.G., J.M.G., J.L.H., J.B.H., C.O.J., J.A.J., C.G.M.K., D.L.K., D.R.K., K.M.K., J.A.K., R.P.K., L.C.K., L.K., H.L., B.R.L., Q.Z.L., M.A.M., J.M., J.M.M., D.R.M., J.T.M., P.M., J.F.M., D.L.M., S.K.N., T.B.N., J.R.O., L.O., N.O., M.P., C.A.P., B.A.P., J.P., P.R., P.S.R., R.R., J.D.R., J.D.R., L.R., L.P.R., J.M.S., C.A.S., J.K.S., L.E.S., H.R.S., R.S., M.F.S., E.D.S., K.L.S., C.S., E.S., A.C.S., S.D.T., S.M.A.T., L.T., B.P.T., T.T., C.V., L.M.V., T.J.V., E.K.W., J.E.W., M.H.W. and D.J.W. revised the manuscript.

## Additional information

**Competing interests:** R.R.G., T.B. and T.W.B. are employees of Genentech, Inc. The remaining authors declare no competing financial interests.

Carl D. Langefeld[1,2], Hannah C. Ainsworth[1,2,*], Deborah S. Cunninghame Graham[3,*], Jennifer A. Kelly[4,*], Mary E. Comeau[1,2], Miranda C. Marion[1,2], Timothy D. Howard[1,5], Paula S. Ramos[6,7], Jennifer A. Croker[8], David L. Morris[3], Johanna K. Sandling[9], Jonas Carlsson Almlöf[9], Eduardo M. Acevedo-Vásquez[10], Graciela S. Alarcón[8], Alejandra M. Babini[11], Vicente Baca[12], Anders A. Bengtsson[13], Guillermo A. Berbotto[14], Marc Bijl[15], Elizabeth E. Brown[8], Hermine I. Brunner[16], Mario H. Cardiel[17], Luis Catoggio[18], Ricard Cervera[19], Jorge M. Cucho-Venegas[10], Solbritt Rantapää Dahlqvist[20], Sandra D'Alfonso[21], Berta Martins Da Silva[22],

Iñigo de la Rúa Figueroa[23], Andrea Doria[24], Jeffrey C. Edberg[8], Emőke Endreffy[25], Jorge A. Esquivel-Valerio[26], Paul R. Fortin[27], Barry I. Freedman[1,28], Johan Frostegård[29], Mercedes A. García[30], Ignacio García de la Torre[31], Gary S. Gilkeson[7], Dafna D. Gladman[32], Iva Gunnarsson[33], Joel M. Guthridge[4], Jennifer L. Huggins[16], Judith A. James[4,34], Cees G.M. Kallenberg[35], Diane L. Kamen[7], David R. Karp[36], Kenneth M. Kaufman[37], Leah C. Kottyan[37], László Kovács[38], Helle Laustrup[39], Bernard R. Lauwerys[40], Quan-Zhen Li[36], Marco A. Maradiaga-Ceceña[41], Javier Martín[42], Joseph M. McCune[43], David R. McWilliams[1,2], Joan T. Merrill[4], Pedro Miranda[44], José F. Moctezuma[45], Swapan K. Nath[4], Timothy B. Niewold[46], Lorena Orozco[47], Norberto Ortego-Centeno[48], Michelle Petri[49], Christian A. Pineau[50], Bernardo A. Pons-Estel[51], Janet Pope[52], Prithvi Raj[36], Rosalind Ramsey-Goldman[53], John D. Reveille[54], Laurie P. Russell[1,2], José M. Sabio[55], Carlos A. Aguilar-Salinas[56], Hugo R. Scherbarth[57], Raffaella Scorza[58], Michael F. Seldin[59], Christopher Sjöwall[60], Elisabet Svenungsson[33], Susan D. Thompson[37], Sergio M.A. Toloza[61], Lennart Truedsson[62], Teresa Tusié-Luna[63], Carlos Vasconcelos[64], Luis M. Vilá[65], Daniel J. Wallace[66], Michael H. Weisman[66], Joan E. Wither[32], Tushar Bhangale[67], Jorge R. Oksenberg[68], John D. Rioux[69], Peter K. Gregersen[70], Ann-Christine Syvänen[9], Lars Rönnblom[71], Lindsey A. Criswell[72], Chaim O. Jacob[73], Kathy L. Sivils[4], Betty P. Tsao[7], Laura E. Schanberg[74], Timothy W. Behrens[67], Earl D. Silverman[75], Marta E. Alarcón-Riquelme[4,76,77], Robert P. Kimberly[8], John B. Harley[37], Edward K. Wakeland[36], Robert R. Graham[67], Patrick M. Gaffney[4] & Timothy J. Vyse[3]

[1]Center for Public Health Genomics, Wake Forest School of Medicine, Winston-Salem, North Carolina 27101, USA. [2]Department of Biostatistical Sciences, Wake Forest School of Medicine, Winston-Salem, North Carolina 27101, USA. [3]Divisions of Genetics and Molecular Medicine and Immunology, Infection and Inflammatory Diseases, King's College London, Guy's Hospital, London SE1 9RT, UK. [4]Arthritis & Clinical Immunology Research Program, Oklahoma Medical Research Foundation, Oklahoma City, Oklahoma 73104, USA. [5]Center for Human Genomics and Personalized Medicine Research, Wake Forest School of Medicine, Winston-Salem, North Carolina 27101, USA. [6]Department of Public Health Sciences, Medical University of South Carolina, Charleston, South Carolina 29425, USA. [7]Department of Medicine, Medical University of South Carolina, Charleston, South Carolina 29425, USA. [8]Division of Clinical Immunology and Rheumatology, UAB School of Medicine, Birmingham, Alabama 35294, USA. [9]Department of Medical Sciences, Molecular Medicine and Science for Life Laboratory, Uppsala University, Uppsala 752 36, Sweden. [10]Departamento de Reumatología, Hospital G. Almenara y Facultad de Medicina, Universidad Nacional Mayor de San Marcos, Lima 15081, Perú. [11]Hospital Italiano de Córdoba, Córdoba X5004BAL, Argentina. [12]Hospital de Pediatría, Centro Médico Nacional Siglo XXI, Instituto Mexicano del Seguro Social, Mexico City 06720, Mexico. [13]Department of Clinical Sciences, Rheumatology, Lund University, Lund 22362, Sweden. [14]Hospital Eva Perón, Granadero Baigorria S2152EDD, Argentina. [15]Department of Internal Medicine and Rheumatology, Martini Hospital, Van Swietenplein 1, 9728, NT, Groningen, The Netherlands. [16]Division of Rheumatology, Department of Pediatrics, Cincinnati Children's Hospital Medical Center and the University of Cincinnati, Cincinnati, Ohio 45229, USA. [17]Centro de Investigación Clínica de Morelia, Morelia, Michoacán 58070, Mexico. [18]Hospital Italiano de Buenos Aires, 1181, Buenos Aires C1181ACH, Argentina. [19]Department of Autoimmune Diseases, Hospital Clínic, University of Barcelona, Barcelona, Catalonia 08007, Spain. [20]Department of Public Health and Clinical Medicine, Division of Rheumatology, Umeå University, Umeå 901 87, Sweden. [21]Department of Health Sciences and Institute of Research in Autoimmune Diseases (IRCAD), University of Eastern Piedmont, Novara 28100, Italy. [22]Unidade Multidisciplinar em Investigação Biomédica/Instituto de Ciências Biomédicas de Abel Salazar—Universidade do Porto, Porto 4099-003, Portugal. [23]Department of Rheumatology, Hospital Universitario de Gran Canaria Dr Negrín, Las Palmas de Gran Canaria 35010, Spain. [24]Division of Rheumatology, Department of Medicine (DIMED), University of Padua, Padua 35122, Italy. [25]Department of Pediatrics and Child Health Center, Albert Szent-Györgyi Medical Center, Faculty of Medicine, University of Szeged, Szeged H-6720, Hungary. [26]Hospital Universitario 'Dr José Eleuterio González' Universidad Autonoma de Nuevo León, Monterrey 64020, México. [27]CHU de Québec Université Laval, Québec, Canada G1R 2JG. [28]Section on Nephrology, Wake Forest School of Medicine, Winston-Salem, North Carolina 27101, USA. [29]Institute of Environmental Medicine, Unit of Immunology and Chronic diseases, Karolinska Institutet, Stockholm 171 77, Sweden. [30]Division of Rheumatology, Hospital Interzonal General de Agudos General San Martín, La Plata 1900, Argentina. [31]University of Guadalajara, Departamento de Fisiología, Guadalajara, Jalisco 44100, Mexico. [32]Centre for Prognosis Studies in The Rheumatic Diseases, Krembil Research Institute, Toronto Western Hospital, Toronto, Ontario M5T 2S8, Canada. [33]Unit of Rheumatology, Department of Medicine Solna, Karolinska Institutet, Karolinska University Hospital, Stockholm SE-171 76, Sweden. [34]Departments of Medicine and Pathology, University of Oklahoma Health Sciences Center, Oklahoma City, Oklahoma 73104, USA. [35]Department of Rheumatology and Clinical Immunology, University Medical Center Groningen, University of Groningen, Groningen 9713 GZ, The Netherlands. [36]Department of Immunology, University of Texas SouthWestern Medical Center, Dallas, Texas 75235, USA. [37]Department of Pediatrics, Center for Autoimmune Genomics and Etiology (CAGE), Cincinnati Children's Hospital Medical Center, Cincinnati, Ohio 45229, USA. [38]Department of Rheumatology, Albert Szent-Györgyi Medical Centre, University of Szeged, Szeged H-6720, Hungary. [39]Department of Rheumatology, Odense University Hospital, Odense 5000, Denmark. [40]Rheumatology, Cliniques Universitaires Saint-Luc & Institut de Recherche Expérimentale et Clinique, Université catholique de Louvain, Louvain-la-Neuve 1348, Belgium. [41]Hospital General de Culiacán, Sinaloa 80220, Mexico. [42]Instituto de Parasitología y Biomedicina López Neyra, CSIC, Granada 18100, Spain. [43]University of Michigan Medical Center, Ann Arbor, Michigan 48103, USA. [44]Centro de Estudios Reumatológicos, Santiago de Chile, Santiago 7500000, Chile. [45]Departamento de Reumatología, Hospital General de México, Mexico D.F., 06726 Mexico. [46]Department of Rheumatology, Mayo Clinic, Rochester, Minnesota 94158, USA. [47]Instituto Nacional de Medicina Genómica (INMEGEN), México City 14610, México. [48]Unidad de Enfermedades Autoimmunes Sistémicas, UGC Medicina Interna, Hospital Universitario San Cecilio, Granada 18007, Spain. [49]Division of Rheumatology, Department of Medicine, Johns Hopkins University School of Medicine, Baltimore, Maryland 21218, USA. [50]Rheumatology Division, McGill University, Montreal, Quebec H3A 0G4, Canada. [51]Department of Rheumatology, Sanatorio Parque, Rosario S2000, Argentina. [52]University of Western Ontario, London, Ontario, Canada M5T 2S8. [53]Division of Rheumatology, Northwestern University Feinberg School of Medicine, Chicago, Illinois 60611, USA. [54]The University of Texas Health Science Center at Houston (UTHealth) Medical School, Houston, Texas 77030, USA. [55]Hospital Universitario Virgen de las Nieves, Granada 18014, Spain. [56]Instituto Nacional de Ciencias Médicas y Nutrición, Department of Endocrinology and Metabolism, Vasco de Quiroga 15, Mexico City 14080, Mexico. [57]Unidad Reumatología y Enfermedades

Autoinmunes H.I.G.A. Dr Alende Mar del Plata, Buenos Aires B7600, Argentina. [58] Referral Center for Systemic Autoimmune Diseases, Fondazione IRCCS Ca'Granda Ospedale Ma Repiore Policlinico and University of Milan, Milan 20122, Italy. [59] Department of Biochemistry and Molecular Medicine, UC Davis School of Medicine, Sacramento, California 95616, USA. [60] Rheumatology Division of Neuro and Inflammation Sciences, Department of Clinical and Experimental Medicine, Linköping University, Linköping 581 83, Sweden. [61] Ministry of Health, San Fernando del Valle de Catamarca, Catamarca K4700, Argentina. [62] Department of Laboratory Medicine, Section of Microbiology, Immunology and Glycobiology, Lund University, Lund 221 00, Sweden. [63] Unidad de Biología Molecular y Medicina Genómica Instituto de Investigaciones Biomédicas/UNAM Instituto Nacional de Ciencias Médicas y Nutrición Salvador Zubirán, Mexico City 14080, Mexico. [64] Hospital Santo Antonio, Universidade do Porto, Porto 4099-003, Portugal. [65] University of Puerto Rico School of Medicine, San Juan 00936, Puerto Rico. [66] Department of Medicine, Cedars Sinai Medical Center, Los Angeles, California 90048, USA. [67] Human Genetics, Genentech Inc, South San Francisco, California 94080, USA. [68] Department of Neurology and Institute of Human Genetics, University of California at San Francisco, San Francisco, California 94158, USA. [69] Université de Montréal and the Montreal Heart Institute, Montreal, Quebec, Canada H1T 1C8. [70] Center for Genomics & Human Genetics, The Feinstein Institute for Medical Research, Manhasset, New York 11030, USA. [71] Department of Medical Sciences, Rheumatology, Uppsala University, 752 36, Sweden. [72] Rosalind Russell/Ephraim P Engleman Rheumatology Research Center, Division of Rheumatology, UCSF School of Medicine, San Francisco, California 94158, USA. [73] Keck School of Medicine of USC, Los Angeles, California 90033, USA. [74] Department of Pediatrics, Duke University, Durham, North Carolina 27708, USA. [75] Department of Pediatrics and the Institute of Medical Sciences, The Hospital for Sick Children, Hospital for Sick Children Research Institute and University of Toronto, Ontario, Canada M5G 1X8. [76] Pfizer-University of Granada-Junta de Andalucía Centre for Genomics and Oncological Research (GENYO), Granada 18007, Spain. [77] Unit of Institute of Environmental Medicine, Karolinska Institute, Solnavägen 171 77, Sweden. * These authors contributed equally to this work.

