## [Peer Review File · Nature Communications]

Reviewer #1 (Remarks to the Author):

This is a well-written and comprehensive manuscript describing a transancestral study of SNP associations in systemic lupus erythematosus cases versus controls from the European, Hispanic American and African populations. SNPs on the immunochip were interrogated, so that limitations of the study pertain to the eurocentricity of variants and the focus on immune genes. Nevertheless, the study provides an important contribution to our understanding of SLE genetics within and between populations.

Comments are below:

One important issue relates to the discussion of population specific risk factors. However, this should be rethought in the light of whether the population is monomorphic for the risk or non-risk allele. This could affect the interpretation of the contribution of many loci in the different populations.

It wasn't clear from reading this manuscript (other than looking at the figures), how many loci had been previously described and how many are novel and this should be made clearer.

The authors describe which alleles are found in chimpanzees (i.e. which are ancestral), but 10% of SNPs that are polymorphic in humans are also polymorphic in chimpanzees – this needs to be considered.

The authors over-interpret a failure to see interactions since they are only looking at SNPs on the immunochip. This needs to be discussed.

Table 2. This is a large table – consider making as supplementary Table.

The authors use evidence for functional relevance from Regulome dB, but none of this is experimentally validated and they should not place too much emphasis on these findings.

Supplementary Table 12: It would be helpful to also present D' values. This section needs to be re-interpreted since the r^2 values for the correlated SNPs are not always convincing.

Reviewer #2 (Remarks to the Author):

1. Although conservative for this array, most ImmunoChip studies use the GWAS significance threshold of $P < 5 \times 10^{-8}$. It is likely that ImmunoChip-wide or less significance levels (tier 2,3,4) produce many false-positive signals that are not replicated in independent cohorts. For example, the authors indicated that all EA Tier 2 associated regions appear novel to EA. I recommend the authors to transfer the detailed results and description about Tier2-4 to Supplementary files.

2. at line 291: SLE-risk HLA alleles were adjusted to identify novel association loci, which is a typical mistake in testing multi-allelic variations in conditional regression analysis. There are many disease-protective HLA alleles with various effect sizes. Their effects are still there. Furthermore, it is very likely that significant contributions of many HLA alleles are not detected in this study due to low statistical power, which does not mean the alleles must be neutral. The authors should adjust for all risk, protective and statistically neutral alleles of the HLA genes (or haplotypes) to completely remove an HLA gene of interest. It is possible that non-HLA SNP rs1150755 (OR=1.33, $P=3.10 \times 10^{-8}$) within TNXB and rs9273448 (OR=0.64, $P=2.39 \times 10^{-8}$) within HLA-DQB1 will be gone after adjusting for all HLA alleles.

3. Some signals are mapped near MHC regions. They should be correctly adjusted for within-MHC signals that do account for entire MHC signals. The novel signals near the MHC region could be detected by cryptic associations with HLA signals. In Figure 2, large amount of residual association signals within the MHC region was shown even after adjusting for HLA signals. The significance levels at UHRF1BP1 are comparable with some signals within the MHC region, which may indicate incorrect controlling for MHC signals.

4. at line 255: "The 8p11 association is not observed in HA or EA" What were the results? Why do the authors think that there were no associations?

5. It seems weird that the authors did not test associations at each amino acid position and residue but analyzed amino acid sequence similarity. Regardless of the type of imputation programs, the dosages of each HLA amino-acid residue can be easily translated from the dosage of imputed four-digit HLA alleles. The association but not the sequence similarity will provide much more explicit

evidence about which positions/residues are critical in the disease pathogenesis. I found that some previous studies already localized some amino acid positions that contribute on risk of SLE.

6. Is there any significant interaction between 0301 and 1501? The authors provided some trends but no statistical evidences for codominance effects. How about the other allele pairs? Why did the authors focus on only two risk alleles? Some homozygotes for a certain allele would be too small to test them. It would be helpful to carry out a comprehensive analysis for common alleles that will give you the subset with higher imputation accuracy and better statistical power (refer to reference #12; Lenz et al.).

Date: February 17, 2017

Re: NCOMMS-16-28510-T

To: Editors and Reviewers,

International
Systemic Lupus
Erythematosus
ImmunoChip
Consortium

We thank the reviewers for their time and thoughts in reviewing our manuscript, *Transancestral mapping and genetic load in systemic lupus erythematosus*. Because of the uniquely large transancestral populations, the level of analytic interrogation, and the excitement in the lupus genetic community about the results, our goal is to make this manuscript a valuable resource for the genetics of lupus and other autoimmune diseases. Your input has been valuable in improving the manuscript. To that end, we have carefully studied your comments and revised the manuscript accordingly (changes in response to reviewers' comments are highlighted in red). Here, we respond to each comment in turn and note any manuscript revisions with a corresponding line number.

Reviewer 1's comments.

1) This is a well-written and comprehensive manuscript describing a transancestral study of SNP associations in systemic lupus erythematosus cases versus controls from the European, Hispanic American and African populations. SNPs on the immunoChip were interrogated, so that limitations of the study pertain to the eurocentricity of variants and the focus on immune genes. Nevertheless, the study provides an important contribution to our understanding of SLE genetics within and between populations.

We thank the reviewer for the encouraging perspective on the value of the consortium's work as reported in this manuscript.

2) One important issue relates to the discussion of population specific risk factors. However, this should be rethought in the light of whether the population is monomorphic for the risk or non-risk allele. This could affect the interpretation of the contribution of many loci in the different populations.

Given the diversity and sample size of our cohorts, we agree on the importance of leveraging allele frequency information to better-classify risk loci as either ancestry dependent or independent. Within the section: "Admixture and population frequencies of SLE-associated SNPs" we aimed to describe and visualize (Figure 4) the ancestral landscape of risk loci. While we explained observations on admixture patterns, we did not properly note that in the entire set of Tier 1 and Tier 2 risk loci, only two loci (in the

discovery ancestral cohort) were nearly monomorphic in the other two ancestral cohorts. One of these loci is on 8p11 and is discussed below. We have updated this section (beginning line 448) and the Figure 4 caption to help clarify this observation to the reader (i.e. drawing their attention to the relative allele frequency in the graphic).

3) *It wasn't clear from reading this manuscript (other than looking at the figures), how many loci had been previously described and how many are novel and this should be made clearer.*

We clarified in the text how the tables highlight what is novel within an ethnicity and across ethnicities. Specifically, Table 1 reports both the overall number of regions and the number of novel regions. Table 2 only reports the novel regions. To delineate this more clearly in the text, we have reworded the single-SNP associations' paragraph to include: 1) list the counts of the novel regions per ethnicity, and 2) emphasize that Table 1 also lists the counts of novel regions (beginning line 241).

4) *The authors describe which alleles are found in chimpanzees (i.e. which are ancestral), but 10% of SNPs that are polymorphic in humans are also polymorphic in chimpanzees – this needs to be considered.*

Given that the actual functional polymorphism is not experimentally determined and we are reporting association of a SNP (effectively tagging a haplotype), and the NCBI ancestral allele is based on the sequence of only one chimp, the reviewer makes an important point. Thus, we have removed the ancestral allele statement (Last sentence in section: "Admixture and population frequencies of SLE-associated SNPs") and its corresponding reference from the manuscript. We have also removed the asterisk indicating ancestral allele status in Supplementary Table 10.

5) *The authors over-interpret a failure to see interactions since they are only looking at SNPs on the immunochip. This needs to be discussed.*

The reviewer's point is well taken, and we have modified the text to clarify. Specifically, we have revised the discussion, beginning on line 519, to emphasize that interactions were limited to ImmunoChip loci. However, even within this more limited context, the observation is important. This study is the only SLE genetics study to date that is reasonably powered for tests of interactions. *Most* ImmunoChip studies published to date have not formally tested interactions, and certainly not in a multi-ancestral sample. Thus, it is important to state that no pairwise-interactions were observed among these major autoimmune disease loci, including with the HLA region. As a consequence it is unlikely, as some have suggested, that statistical interactions explain a significant proportion of the "missing heritability" and risk for SLE.

6) *Table 2. This is a large table – consider making as supplementary Table.*

Table 2 explicitly lists the novel loci (novel to SLE and novel to that specific ethnicity). Because of this, we feel it is important to retain this admittedly large table as a primary table and resource. Given that Nature Communications is an online journal, we

believe it is important to place this major inference in the body of the paper and not in the supplemental tables. We also note that due to the size of Table 2, we have limited our main display figures to 9 (tables and figures), versus the allowed maximum of 10.

7) (R1; C6) The authors use evidence for functional relevance from Regulome dB, but none of this is experimentally validated and they should not place too much emphasis on these findings.

The reviewer is correct that this manuscript does not contain experimentally validated functional inferences. However, RegulomeDB and HaploReg allow us to consider what is known about the regions and generates hypotheses that individual investigators can test. To that end, we only used the RegulomeDB and HaploReg to annotate the results and provide potential context for follow-up studies. We note that it is viewed as standard practice to report such results, and we would be criticized by some readers for its omission. Thus, we have retained the hypothesized functional information but addressed this reviewer's concern by including a clarifying reminder that these results require robust experimental validation (line 486).

8) Supplementary Table 12: It would be helpful to also present D' values. This section needs to be re-interpreted since the r² values for the correlated SNPs are not always convincing.

We have considered the reviewer's comment carefully. The statistic D' really is less appropriate for tests of association. Specifically, a D'=1 between two SNPs merely means that there is a deficit of haplotypes (e.g., three instead of four); a valuable measure if one is interested in linkage disequilibrium patterns in the genome as they might vary across subpopulations or ancestral groups. However, the corresponding association tests for the two SNPs do not have to be strongly correlated, even if D'=1. In contrast, the R² statistic is defined as the square of the correlation for the corresponding 2x2 allele table, and highly correlated SNPs as defined by the R² statistic mathematically result in similar levels of association. An R²=0.50 (the threshold used here) can be interpreted as the proportion of variation for one SNP that is explained by the other. We sympathize with the reviewer's point that as the R² approaches 0.50, it may be less "convincing" but it is suggestive and motivates hypotheses that can be experimentally tested.

Reviewer 2's comments.

1) Although conservative for this array, most ImmunoChip studies use the GWAS significance threshold of P<5×10⁻⁸. It is likely that ImmunoChip-wide or less significance levels (tier 2,3,4) produce many false-positive signals that are not replicated in independent cohorts. For example, the authors indicated that all EA Tier 2 associated regions appear novel to EA. I recommend the authors to transfer the detailed results and description about Tier2-4 to Supplementary files.

We have only included the novel loci of Tier 1 and Tier 2 in the main manuscript (loci that were not novel to SLE or to the ethnicity were kept as supplementary information). We felt that even including all Tier 1 SNPs ($p < 5.0 \times 10^{-8}$) is too large of a table for the body of the manuscript. The Tier 2 criterion ($p < 1.0 \times 10^{-6}$) is even more conservative than the formal Bonferroni criteria; the

Bonferroni method of multiple comparison adjustment is known to be too conservative. The identification of so many novel loci and corroborating known loci are functions of the exceptional power of the sample size and the *a priori* nature of the regions selected for the ImmunoChip; SNP content of the ImmunoChip was based on empirical evidence from several rheumatic disease researcher groups. Further, the more liberal false discovery rate (FDR) threshold, still an established multiple comparison procedure, is restricted to supplemental material. We gently note that there is no Tier 4 criterion. Although we respect the reviewer's concern and recognize the need to balance type 1 and type 2 errors, we believe our approach is appropriate and have added clarification for these thresholds in the manuscript (line 241).

2) At line 291: SLE-risk HLA alleles were adjusted to identify novel association loci, which is a typical mistake in testing multi-allelic variations in conditional regression analysis. There are many disease-protective HLA alleles with various effect sizes. Their effects are still there. Furthermore, it is very likely that significant contributions of many HLA alleles are not detected in this study due to low statistical power, which does not mean the alleles must be neutral. The authors should adjust for all risk, protective and statistically neutral alleles of the HLA genes (or haplotypes) to completely remove an HLA gene of interest. It is possible that non-HLA SNP rs1150755 (OR=1.33, P=3.10x10⁻⁸) within TNXB and rs9273448 (OR=0.64, P=2.39x10⁻⁸) within HLA-DQB1 will be gone after adjusting for all HLA alleles.

In the section mentioned by the reviewer, *SNP associations after adjusting for HLA alleles*, there was a descriptive inaccuracy: “*after adjusting for HLA risk alleles*”. We thank the reviewer for catching this misnomer. The alleles used to adjust for HLA effects were from the multi-locus model mentioned in the previous section. This model included both risk and non-risk alleles. We have updated the section (beginning at line 291) to appropriately describe this: “... *adjusting for HLA alleles (risk and protective) identified in the stepwise modeling.*” We agree with the reviewer that there is the potential for multiple HLA alleles to exhibit various effects on SLE-status. Further, a lack of detecting statistical differences is not the same as showing that the SNPs are neutral – a probabilistic interpretation of the latter requires formal equivalence testing while adjusting for the linkage disequilibrium in the region. While we recognize the reviewer's suggestion to include all risk, protective, and neutral alleles, including the ~100+ alleles (many rare and in high linkage disequilibrium), such an approach would not produce statistically robust models. In an attempt to capture alleles contributing smaller effects, our stepwise modeling approach included an entry and exit criteria of $p \leq 0.01$ and **all alleles** passing quality control (risk or protective and regardless of single-allele associations) were used as input (to be considered for entry/exit to the model). We computed a power analysis to understand what we might have failed to capture due to a modest effect size of an HLA allele. This study had ~0.80 power to detect odds ratios of 1.16, 1.13, 1.11 and 1.10 for reference allele frequencies of 0.05, 0.10, 0.13, and 0.15, respectively. Although not neutral, the combination of effect size and allele frequency indicates that these loci would have modest contribution to risk and are not likely to affect detection of other loci. We note that after adjusting for SLE-associated HLA alleles (across more than 5 Mb of the HLA region) only two regions of SNP associations that met Tier 1 or Tier 2 threshold were observed in the EA cohort (rs1150755 and rs9273448). The SNPs that were associated after adjusting for the HLA alleles had larger effect sizes than discussed in the power analysis. Thus, we believe our interpretation is more parsimonious. That is, we believe it is more likely that these two regions (SNPs) are either HLA associations not captured by the classic HLA alleles

imputed here or even more likely are the results of imputation imprecision. To correct the error discovered by the reviewer, we have revised the manuscript to clearly state that we are considering both risk and protective alleles (beginning at line 291).

3) Some signals are mapped near MHC regions. They should be correctly adjusted for within-MHC signals that do account for entire MHC signals. The novel signals near the MHC region could be detected by cryptic associations with HLA signals. In Figure 2, large amount of residual association signals within the MHC region was shown even after adjusting for HLA signals. The significance levels at UHRF1BP1 are comparable with some signals within the MHC region, which may indicate incorrect controlling for MHC signals.

We apologize if our approach was not clear. As noted above, the stepwise modeling included all alleles that met a $p \leq 0.01$ (risk or protective) across the extended MHC. Thus, even smaller effect sizes are accounted for in the modeling. The associations in the UHRF1BP1 area noted by the reviewer are adjusted via the logistic regression for these risk and protective alleles meeting the stepwise significance of $p \leq 0.01$. To clarify the results, we have now emphasized in the caption of Figure 2 that the axis scale is changed between the top and bottom plots (not-adjusted for HLA versus adjusted for HLA alleles, respectively) by noting that the red line identifies the same Tier 1 ($p < 5.0 \times 10^{-8}$) threshold in each plot. We applied the HLA-adjusted model to the extended MHC and beyond, including the UHRF1BP1, to identify associations masked or artificially enhanced by SLE-associated HLA alleles. The region surrounding UHRF1BP1 is beyond the extended MHC region and was found to not be influenced by the inclusion of HLA alleles. Linkage disequilibrium patterns confirm this independence (please see plot below).

Response Letter Figure 1. None of the SNPs across the MHC appear to be in linkage disequilibrium with our TOP associated SNP in UHRF1BP1

EA LD Across MHC

4) at line 255: “The 8p11 association is not observed in HA or EA” What were the results? Why do the authors think that there were no associations?

In considering the reviewer’s comment, we realized we omitted information that would be helpful to the reader. The 8p11 locus was one of two (Tier 1 and Tier 2) SNPs that was found to be effectively monomorphic in the other (non-discovery) ancestries. Specifically, of the 193 Tier 1 and 2 SNPs, only two loci (rs1804182 AA Tier 1 and rs11845506 HA Tier 2) were nearly monomorphic (frequency < 0.003) in the other ancestral cohorts. This suggests that the vast majority of the ancestry-specific SNP associations were not driven by the presence of monomorphic alleles in the other two (non-discovery) cohorts. These allele patterns are further illustrated in Figure 4. To correct for this omission, we have included a statement on the general lack of monomorphism in other ethnicities and reference the exceptions noted above (line 261).

5) It seems weird that the authors did not test associations at each amino acid position and residue but analyzed amino acid sequence similarity. Regardless of the type of imputation programs, the dosages of each HLA amino-acid residue can be easily translated from the dosage of imputed four-digit HLA alleles. The association but not the sequence similarity will provide much more explicit evidence about which positions/residues are critical in the disease pathogenesis. I found that some previous studies already localized some amino acid positions that contribute on risk of SLE.

While we recognize that a number of recent studies have tested for associations among phenotypes and individual amino acid residues, we and others believe this type of analysis is not the most informative and may be misleading.

Importantly, proteins are not linear structures; they exist in three-dimensional configurations which mediate their functionality and interactions (i.e., with DNA or other proteins). Thus, it has only been in understanding the three-dimensional structures that studies have begun to appreciate the high-order interactions among amino acids in a protein's sequence (e.g., upon solving the p53 protein structure, it was finally realized that the seemingly un-connected amino acid mutations were centralized at the protein-DNA interface). Amino acids do not act independently; their hydrophobic, geometric, and bonding conformations all contribute to their surrounding environments.

The MHC is among the most polymorphic regions in the genome and this translates to HLA molecules with high variability across their amino acid sequences. For most proteins such variability would lead to deleterious functional changes; however, the variability in HLA molecules is evolutionarily advantageous, providing recognition of a wide variety of epitopes. This variability introduces analytic challenges. While non-HLA protein analyses may only need to focus on a relatively small number of mutations, the HLA molecules exhibit an exponentially greater degree of variability, with each mutation posing a potential effect on the molecule's functionality (epitope binding, T-cell binding, cell-surface stability). This complexity was well-established by the 1994 crystal structure of HLA-DR1 (Stern et al; *Nature*). Here, they described the great degree of variability in the binding pockets and how even underlying (not exposed to the pocket's surface) amino acids influenced epitope and T-cell binding. These findings have been further established through epitope studies, such as Hemmer et al. (*J. of Imm.*; 2000) where epitope interactions were found to be mediated through non-additive combinations of amino acid mutations. Additionally, while there are 'anchoring residues' in the peptide binding pocket, even their contributions are influenced by surrounding amino acids.

From a statistical perspective, Segal et al., (2001; *Biometrics*) dissected the shortcomings of analyses that failed to incorporate the higher-order interactions of amino acids in HLA molecules. Specifically, they pointed out that standard regression techniques were simply unable to capture the numerous unordered categorical covariates introduced by interactions in the highly polymorphic HLA molecules (~2,451 terms to capture all possible amino acid substitutions by ~10k terms to account for all second-order interactions –note: this does not even include third order interactions).

The ability to impute 4-digit HLA alleles also provides the ability to impute amino acid dosages and evaluate these amino acids (for the first time) in large-scale cohorts. While it is important to leverage and utilize new methods to thoroughly investigate datasets, the number of samples in these studies does not negate the previously mentioned analytic challenges. So while such studies have identified specific amino acids using standard regression techniques (which, again, do not account for higher-order interactions), the biological relevance of these findings requires investigation. Van Heemst et. al., (2015; *Curr Opin Rheumatol.*) pointed out that nearly every study that has used this technique has identified the same two amino acids (positions 11 and 13 of the

HLA-DR Beta Chain). While it is certainly possible that these are widely-important positions, they also correspond to the most variable amino acid positions. Thus, studies claiming that the majority of MHC association can be attributed to these positions might instead be identifying positions that simply tag/differentiate specific HLA genotypes; and so the biological relevance of these findings requires further investigation (Van Der Woude et al. 2015; Best Practice & Research Clinical Rheumatology).

To elucidate the importance of amino acids in HLA molecules, clustering and machine learning (non-linear) techniques are suggested (Segal et al., 2001). Many studies utilize HLA supertypes which classify HLA alleles into subtypes by functional observations (i.e. epitope binding; motif patterns); however, strategies such as epitope binding are not without potential faults, such as database-bias. Here, we clustered solely by sequence similarity and then evaluated SLE-associations among clusters of 95% sequence identity. This was used as a means of limiting the number of variables when comparing across HLA alleles. Using this method we identified several patterns that may be informative for follow-up functional studies: lack of overlap of similarity between the two primary SLE-risk haplotypes (DR3 and DR15) and identification of a unique biochemical characteristic of the DR15 risk alleles (hydrophobic residues in pocket 7) in comparison the non-risk alleles with highest sequence similarity. This approach forms one of the important novel components of the paper.

In conclusion, we recognize many recent studies have computed association analyses at the amino acid-level; however, given biological and statistical considerations, we do not believe this type of analyses to be the most informative and may mislead researchers.

6) Is there any significant interaction between 0301 and 1501? The authors provided some trends but no statistical evidences for codominance effects. How about the other allele pairs? Why did the authors focus on only two risk alleles? Some homozygotes for a certain allele would be too small to test them. It would be helpful to carry out a comprehensive analysis for common alleles that will give you the subset with higher imputation accuracy and better statistical power (refer to reference #12; Lenz et al.).

After reading the reviewer's comment and reviewing what we had previously written, we recognized that we did not sufficiently motivate the comparisons presented. We have expanded this section to address the reviewer's concerns. In SLE, there are two primary risk haplotypes (DR3-DQ2 and DR15-DQ6) which are comprised of alleles in strong linkage disequilibrium. Thus, we selected DRB1*03:01 and DR*15 (DRB1*15:01 in EA & HA; DRB1*15:03 in AA) as tagging alleles to evaluate risk allele heterozygosity. We note that these alleles are common and provide reasonable power for interaction analysis. We did not observe conclusive evidence of an interaction for people having both of these risk alleles. However, the lack of fit test (see Methods) strongly supported the dominance model of risk (departure from additivity) for individual DR3 and DR15 SLE risk alleles. These results are now included (line 318). We acknowledge that there is the opportunity for a more in-depth analysis of HLA interactions but this is beyond the scope and intent of this manuscript. Therefore, we plan to pursue this in an independent manuscript.

Reviewer #1 (Remarks to the Author)

The authors have responded appropriately and I have no other concerns.

Reviewer #2 (Remarks to the Author)

This reviewer respects the author's responses and opinion.